# Optimal Reinsurance under the Linear Combination of Risk Measures in the Presence of Reinsurance Loss Limit

**Qian Xiong [1], Zuoxiang Peng [1] and Saralees Nadarajah [2],***

[1] School of Mathematics and Statistics, Southwest University, Chongqing 400715, China; xiongqian2018@126.com (Q.X.); pzx@swu.edu.cn (Z.P.)

[2] Department of Mathematics, University of Manchester, Manchester M13 9PL, UK

* Correspondence: mbbsssn2@manchester.ac.uk

**Abstract:** Optimal reinsurance problems under the risk measures, such as Value-at-Risk (VaR) and Tail-Value-at-Risk (TVaR), have been studied in recent literature. However, losses based on VaR may be underestimated and TVaR allows us to account better for catastrophic losses. In this paper, we propose a new family of flexible risk measures denoted by LVaR, which is a weighted combination of VaR and TVaR. Based on the new risk measures, we deal with the optimal reinsurance problem by minimizing the LVaR of the total risks of an insurer when two types of constraints for reinsurer's risk exposure are considered. The results indicate that the two-layer reinsurance is always an optimal reinsurance policy with both types of constraints. Also, we find that the optimal reinsurance policy depends on the confidence level, the weight coefficient, the safety loading, the tolerance level, as well as the relations between them. Finally, we illustrate the results by numerical examples and compare them with the results in Lu et al.

**Keywords:** expected value principle; loss limit constraint; optimal reinsurance; tail-value-at-risk; value-at-risk

## 1. Introduction

Reinsurance can be seen as a risk transfer that may help to reduce the risk exposure of the insurer and, hence, to stabilize the business. More precisely, the insurer transfers some part of risk to a reinsurance company at the expense of paying the corresponding reinsurance premium, which reduces the potential risk. Thanks to this tool, catastrophic risk, such as climate risks, initially hardly insurable, may become insurable by transferring risks (see, e.g., Charpentier 2008). Naturally, the reinsurance premiums increase as the risk transferred to a reinsurer increases. In this context, how to deal with the trade-off between the risk retained and the premium paid to the reinsurer becomes a major issue for an insurer.

The study of the optimal reinsurance problems can be traced back to the seminal papers of Borch (1960) and Arrow (1963). Calculating the reinsurance premium by using the expected value principle, Borch (1960) showed that the stop-loss reinsurance is optimal when the objective is to minimize the variance of the insurer's retained loss. Choosing the same premium principle as Borch (1960), Arrow (1963) considered the optimal reinsurance problem by maximizing expected utility of the terminal wealth of a risk-averse insurer. The results also showed that the stop-loss reinsurance is optimal. Kaluszka (2001) extended Borch's result by applying the mean-variance premium principle. Arrow's result has been considered under other premium principles, see Young (1999) and Kaluszka (2005). For different purposes, the optimal reinsurance problem has been studied under more intricate optimization criteria and/or more general premium principles, see Albrecher et al. (2017) for a survey, and some examples. When the reinsurance premium is calculated according to the maximal possible claims principle, Kaluszka and Okolewski (2008) showed that the limited stop-loss and the truncated stop-loss are the optimal reinsurance contracts

by maximizing the expected utility. Guerra and Centeno (2010) considered the optimal forms of reinsurance when the insurer seeks to maximize the adjustment coefficient of the retained risk. Using the maximization of the expected utility of terminal wealth, Zhang and Siu (2012) and Liang and Bayraktar (2014) studied the optimal reinsurance and investment strategies with different market assumptions.

In the fields of finance and insurance, Value-at-Risk (VaR) and Tail-Value-at-Risk (TVaR) are the most popular risk measures due to their merits and good properties. For this reason, many optimal reinsurance problems under the VaR and/or the TVaR have been studied in the literature. For instance, under the expected value premium principle, Cai and Tan (2007) provided the optimal retention of a stop-loss reinsurance by minimizing the VaR and the TVaR of the insurer's total risk exposure. Cai et al. (2008) derived the optimal reinsurance within a class of increasing convex loss functions. Cheung (2010) extended the VaR-minimization reinsurance model in Cai et al. (2008) by considering Wang's premium principle. Lu et al. (2016) considered the optimal reinsurance problem under the optimality criteria of VaR and TVaR risk measures when constraints for the reinsurer's risk exposure are presented. Further extensions of the optimal reinsurance problems can be found in Balbás et al. (2009), Tan et al. (2011), and Chi and Tan (2011, 2013).

Let $X$ denote the amount of loss initially assumed by an insurer in a given time period. Usually, $X$ is assumed to be a non-negative random variable defined on the probability space $(\Omega, \mathcal{F}, \mathbb{P})$ with continuous distribution function $F_X(x)$. We further assume that $F_X(x)$ is strictly increasing on $(0, +\infty)$ but with a possible jump at 0 with $\mathbb{P}(X = 0) = p_0$. $S_X(x) = 1 - F_X(x)$ and $f_X(x)$ are used to denote the survival function and the density function of $X$, respectively. Given a confidence level $\alpha \in (0, 1)$, the VaR and the TVaR of the risk $X$ are defined as

$$\text{VaR}_\alpha(X) = \inf\{x \in \mathbb{R} : F_X(x) \geq \alpha\} \tag{1}$$

and

$$\text{TVaR}_\alpha(X) = \frac{1}{1 - \alpha} \int_\alpha^1 \text{VaR}_s(X) ds, \tag{2}$$

respectively. For a continuous loss distribution, there are many alternative names of TVaR, such as Expected Shortfall (ES), Conditional Tail Expectation (CTE), Average VaR (AVaR), and Conditional VaR (CVaR). We refer to Acerbi and Tasche (2002) and Rockafellar and Uryasev (2013) for the relationships between the various notions. It is well known that VaR is simple and easier to interpret, but it is, in general, not subadditive and, hence, not coherent in the sense of Artzner et al. (1999). Another major drawback of VaR is that it only takes into account the probability of a big loss, not the size of the loss, i.e., it completely ignores the tail loss beyond the reference VaR. As a result, the same VaRs may occur when dealing with different extreme losses, and moreover, VaR may underestimate the losses in practice, especially when heavy-tailed losses are incorrectly modeled with light-tailed distributions, such as the normal distribution. By comparison with VaR, TVaR defines a more conservative risk measure that is always subadditive. Since it is interpreted as the arithmetic average of $\text{VaR}_\alpha$ over all levels $s \geq \alpha$, capital reserves based on TVaR are always larger than those based on VaR. Further discussions about the comparative advantages of TVaR and VaR, we refer to Embrechts and Hofert (2014), McNeil et al. (2015), and references therein.

As a generalization of VaR, the spectral risk measure has received attention in recent years. In fact, the spectral risk measure is a weighted average of $\text{VaR}_\alpha$ with the weight function depending on the user's risk-aversion, so that it could help us to link the risk measure to the user's attitude towards risk ((Acerbi 2002) and Dowd et al. (2008)). Another interesting generalization of $\text{VaR}_\alpha$ is the Lambda Value-at-Risk, which considers the dependence between the level $\alpha$ and the amount of the loss (see, e.g., Frittelli et al. 2014; Bellini and Peri 2022). Recall that, in the risk-management community, there is an ongoing debate

on the advantages and disadvantages of TVaR and VaR. However, there is no evidence for global advantage of one risk measure against the other, and the size of capital reserves may be significantly different depending on which risk measure is used. To provide a risk assessment between that offered by TVaR and VaR, and capture more information about various attitudes towards risk, in this paper, we extend VaR and TVaR to a more general family of risk measures, denoted by LVaR, which is a linear combination of VaR and TVaR. Obviously, LVaR includes VaR and TVaR as special cases and allows us to consider VaR and TVaR simultaneously.

As shown in the literature, once the optimization criteria and the premium principle are determined, the optimal reinsurance problem becomes a purely mathematical problem, which makes the analysis easier. The expected value principle is one of the most popular premium principles both due to its transparency and simplicity (see, e.g., Albrecher et al. 2017), so that we assume that the reinsurance premium is calculated by this premium principle in this paper. To determine the amount of risk retained (or the premium paid to the reinsurer), the insurer has to make a choice among all feasible reinsurance treaties, and a reasonable criterion for the insurer is naturally to choose the reinsurance form which makes its total risk to be as small as possible. To this end, we revisit the optimal reinsurance problem by using the new risk measure LVaR to quantify the total risk exposure of the insurer.

Recently, attention has been paid to controlling reinsurer's risk since the insurer may be under a heavy financial burden with no limit on coverage. Throughout the paper, we will consider two types of constraints proposed by Cummins and Mahul (2004) and Zhou et al. (2010). The first type due to Cummins and Mahul (2004) has the ceded losses constrained to be less than the upper limit. The second type is that the reinsurer's loss after the payments is limited to be less than a certain predetermined level. As mentioned above, the motivation behind these two classes of upper limits on coverage arises from providing the insurer with limited liability with respect to the indemnity schedule. Instead of constraining the risk of the reinsurer, Boonen et al. (2016) established lower and upper bounds of the reinsurance premium which ensure the benefits of reinsurance to both insurers and reinsurers. More recently, their work has been generalized by Boonen et al. (2021) to the case where an insurer could use more than one reinsurer to reinsure its risk, i.e., the case where there is competition among multiple reinsurers. With similar constraints, Balbás et al. (2022) showed that the optimal reinsurance problem may be very complex if the expected profits of both insurers and reinsurers are required to be non-decreasing when the reinsurance contract is signed. Here, we focus on the two types of constraints discussed in Cummins and Mahul (2004) and Zhou et al. (2010).

Many researchers have discussed the existence of optimal reinsurance contracts. However, closed-form expressions of the optimal risk transferred to the reinsurer and the resulting total risk of the insurer have not been provided. Furthermore, the study of the optimal reinsurance problems under the VaR and the TVaR is generally discussed separately, which ignores the situation where one may consider both VaR and TVaR simultaneously. Thus, inspired by Lu et al. (2016), we derive the optimal transferred risk in closed-form, but the most important distinction is that we seek to deal with the optimal reinsurance problem under the LVaR, while Lu et al. (2016) considered this problem under the VaR and the TVaR separately. To sum up, the main contributions of the present paper are as follows. First, we introduce a new family of risk measures LVaR to capture more information about various attitudes towards risk and to enable us to consider VaR and TVaR simultaneously. Second, by minimizing LVaR of the insurer's total risk and using the expected value principle, we identify optimal reinsurance contracts when the reinsurer's risk exposure is constrained. It appears that the two-layer reinsurance is always an optimal reinsurance policy which shows the stability of optimality results when switching from VaR and TVaR to LVaR. Moreover, we show that the solutions of optimal reinsurance model in Lu et al. (2016) can be unified and generalized by using the LVaR. In addition, the optimal quantity of ceded losses depends on the confidence level $\alpha$, the weight coefficient $\omega$, the safety loading $\theta$, the

tolerance level $L$ (or $K$), as well as the relations between them. For the insurer, our results provide explicit expressions of the optimal reinsurance contract which may help to foster the intuition and comprehension of the consequences of setting up a reinsurance contract.

The rest of the paper is organized as follows. In Section 2, we give some preliminaries, define a new family of risk measures, and describe the setup of the proposed reinsurance models. Section 3 states the main results and is structured as follows. Section 3.1 studies the optimal reinsurance problem under the new risk measures optimality criterion by considering the first type of constraint. Section 3.2 studies the same problem in Section 3.1 with the second type of constraint. Section 4 illustrates the results in Section 3 by numerical examples and also compares them with the results in Lu et al. (2016). Section 5 concludes our study. All proofs are given in the Appendix A.

## 2. Preliminaries

### 2.1. A New Risk Measure

Recall that, from the definitions in (1) and (2), the VaR measures the minimum loss, so that a disadvantage when using VaR is that catastrophic losses can be underestimated. Since the TVaR measures average losses in the most adverse cases rather than just the minimum loss, naturally, losses based on TVaR are much larger than those based on VaR. Therefore, our objective is to propose a new family of risk measures which provides a risk assessment that lies somewhere between those offered by the VaR and the TVaR. The new family of risk measures, named LVaR, is defined as a linear combination of VaR and TVaR, and is helpful in providing risk managers with more flexible tools. We now formally define LVaR.

**Definition 1.** *The* LVaR *of a random variable $X$ at a confidence level $\alpha \in (0,1)$ is defined as*

$$\text{LVaR}_\alpha(X) = \omega \cdot \text{TVaR}_\alpha(X) + (1-\omega) \cdot \text{VaR}_\alpha(X),$$

*where $\omega \in [0,1]$ is the weight coefficient.*

**Remark 1.** *Obviously, the* LVaR *risk measures include* VaR *and* TVaR *as special cases by setting $\omega = 0$ and $\omega = 1$, respectively. Moreover, we can easily check that $\text{VaR}_\alpha(X) \le \text{LVaR}_\alpha(X) \le \text{TVaR}_\alpha(X)$.*

**Remark 2.** *Note that different perspectives will lead to different weight coefficients. For example, a more pessimistic insurer may prefer the larger $\omega$. As a consequence, the choice of the weight coefficient $\omega$ can be complicated in practice, and hence is not addressed here.*

### 2.2. Optimal Reinsurance Model

Under a typical reinsurance arrangement, let $I(X)$ be the part of loss transferred from the insurer to a reinsurer which we refer to as the ceded loss function. As usual in the reinsurance policy, we assume that the ceded loss function $I(X)$ satisfies the following properties: (1) $I(0) = 0$ and $I(x)$ is an increasing function, and (2) $I(x_2) - I(x_1) \le x_2 - x_1$ for any $0 \le x_1 \le x_2$. It is easy to check that these conditions imply that $0 \le I(x) \le x$ for any $x \ge 0$. In the sequel, we will denote by $\mathscr{F}$ the set of all ceded loss functions satisfying conditions (1) and (2). Associated with the ceded loss function $I(X)$, we denote $R_I(X) = X - I(X)$ as the retained loss of the insurer. As an exchange of the undertaking risk, the insurer should pay a reinsurance premium to the reinsurer. Here, we assume that the reinsurance premium is determined by the common expectation premium principle and expressed as

$$\Pi_I(X) = (1+\theta)\,\mathbb{E}[I(X)],$$

where $\theta > 0$ is the safety loading factor. Consequently, the total risk exposure of the insurer in the presence of reinsurance, denoted by $T_I(X)$, is the sum of the retained loss and the reinsurance premium, i.e.,

$$T_I(X) = R_I(X) + \Pi_I(X).$$

The objective of this paper is, by minimizing the new risk measure LVaR, to consider the optimal reinsurance problem when the constraints for the reinsurer's risk exposure are presented. Two types of constraints due to Cummins and Mahul (2004) and Zhou et al. (2010) are considered. Specifically, we are interested in seeking an optimal reinsurance policy over the following two sets of ceded loss functions:

- $\mathscr{F}_1 = \{I(x) \in \mathscr{F} \,|\, I(x) \leq L\}$;
- $\mathscr{F}_2 = \{I(x) \in \mathscr{F} \,|\, I(x) - \Pi_I(X) \leq K\}$.

For $I \in \mathscr{F}_i$, $i = 1, 2$, $\mathrm{LVaR}_\alpha(T_I(X))$ can be seen as a function of $I(X)$, and it suggests the amount of assets required to protect against adverse outcomes of the total risk $T_I(X)$. For a fixed confidence level, a prudent risk management generally requires $\mathrm{LVaR}_\alpha(T_I(X))$ to be as small as possible. This allows us to determine the optimal ceded loss function by minimizing $\mathrm{LVaR}_\alpha(T_I(X))$ in the above classes of ceded loss functions. Our optimal reinsurance model can be formulated as follows:

LVaR **optimization criterion**

$$\mathrm{LVaR}_\alpha(T_{I^*}(X)) = \min_{I \in \mathscr{F}_i} \{\mathrm{LVaR}_\alpha(T_I(X))\}, \tag{3}$$

where $i = 1, 2$, and $I^*$ is the resulting optimal ceded loss function.

We conclude this section by introducing the following notations:

$$\delta = 1 + \theta - \frac{\omega}{1 - \alpha}, \qquad \theta^* = \frac{\theta}{1 + \theta}.$$

## 3. Main Results

*3.1. Optimal Reinsurance under the Constraint of $\mathscr{F}_1$*

In this section, we will solve (3) under the constraint $\mathscr{F}_1$:

$$\mathrm{LVaR}_\alpha(T_{I^*}(X)) = \min_{I \in \mathscr{F}_1} \{\mathrm{LVaR}_\alpha(T_I(X))\}. \tag{4}$$

First, note that both VaR and TVaR are translation invariant and

$$\mathrm{VaR}_\alpha(g(X)) = g(\mathrm{VaR}_\alpha(X)) \tag{5}$$

for any increasing and continuous function $g$ (see McNeil et al. (2015) and Theorem 1 in Dhaene et al. (2002)). Thus,

$$
\begin{aligned}
\mathrm{VaR}_\alpha(T_I(X)) &= \mathrm{VaR}_\alpha(R_I(X) + \Pi_I(X)) \\
&= \mathrm{VaR}_\alpha(R_I(X)) + \Pi_I(X) \\
&= R_I(\mathrm{VaR}_\alpha(X)) + \Pi_I(X) \\
&= \mathrm{VaR}_\alpha(X) - I(\mathrm{VaR}_\alpha(X)) + (1 + \theta)\,\mathbb{E}[I(X)]. 
\end{aligned} \tag{6}
$$

Moreover, by noticing that $\mathrm{TVaR}_\alpha(X) = \mathrm{VaR}_\alpha(X) + \frac{\int_{\mathrm{VaR}_\alpha(X)}^\infty S_X(x)dx}{1 - \alpha}$, we can obtain that

$$
\begin{aligned}
&\mathrm{TVaR}_\alpha(T_I(X)) \\
={}& \mathrm{TVaR}_\alpha(R_I(X) + \Pi_I(X)) \\
={}& \mathrm{TVaR}_\alpha(R_I(X)) + \Pi_I(X) \\
={}& \mathbb{E}[R_I(X) | R_I(X) > \mathrm{VaR}_\alpha(R_I(X))] + \Pi_I(X) \\
={}& \mathbb{E}[R_I(X) - \mathrm{VaR}_\alpha(R_I(X)) | R_I(X) > \mathrm{VaR}_\alpha(R_I(X))] + \mathrm{VaR}_\alpha(R_I(X)) + \Pi_I(X) \\
={}& \frac{\int_{\mathrm{VaR}_\alpha(R_I(X))}^\infty [R_I(x) - \mathrm{VaR}_\alpha(R_I(X))]dF_{R_I(X)}(x)}{\mathbb{P}(R_I(X) > \mathrm{VaR}_\alpha(R_I(X)))} + \mathrm{VaR}_\alpha(R_I(X)) + \Pi_I(X)
\end{aligned}
$$

$$= \quad \frac{1}{1-\alpha} \mathbb{E}\big[(R_I(x) - \text{VaR}_\alpha(R_I(X)))_+\big] + \text{VaR}_\alpha(X) - I(\text{VaR}_\alpha(X)) + (1+\theta)\,\mathbb{E}[I(X)]. \tag{7}$$

By applying (5) again, we have

$$
\begin{aligned}
& \mathbb{E}\big[(R_I(x) - \text{VaR}_\alpha(R_I(X)))_+\big] \\
=\ & \mathbb{E}\big[(X - I(X) - \text{VaR}_\alpha(X) + I(\text{VaR}_\alpha(X)))_+\big] \\
=\ & \mathbb{E}\Big[(X - I(X) - \text{VaR}_\alpha(X) + I(\text{VaR}_\alpha(X)))_+ \big(\mathbb{I}_{\{X \geq \text{VaR}_\alpha(X)\}} + \mathbb{I}_{\{X < \text{VaR}_\alpha(X)\}}\big)\Big] \\
=\ & \mathbb{E}\Big[(X - I(X) - \text{VaR}_\alpha(X) + I(\text{VaR}_\alpha(X))) \cdot \mathbb{I}_{\{X \geq \text{VaR}_\alpha(X)\}}\Big] \\
=\ & \mathbb{E}\Big[(X - \text{VaR}_\alpha(X)) \cdot \mathbb{I}_{\{X \geq \text{VaR}_\alpha(X)\}}\Big] - \mathbb{E}\Big[(I(X) - I(\text{VaR}_\alpha(X))) \cdot \mathbb{I}_{\{X \geq \text{VaR}_\alpha(X)\}}\Big] \\
=\ & \mathbb{E}\big[(X - \text{VaR}_\alpha(X))_+\big] - \int_{\text{VaR}_\alpha(X)}^{\infty} I(x) - I(\text{VaR}_\alpha(X))dF_X(x) \\
=\ & \mathbb{E}\big[(X - \text{VaR}_\alpha(X))_+\big] - \int_{\text{VaR}_\alpha(X)}^{\infty} I(x)dF_X(x) + (1-\alpha)I(\text{VaR}_\alpha(X)). \tag{8}
\end{aligned}
$$

Substituting (8) into (7), we obtain

$$\text{TVaR}_\alpha(T_I(X)) = \frac{1}{1-\alpha} \mathbb{E}\big[(X - \text{VaR}_\alpha(X))_+\big] + \text{VaR}_\alpha(X) - \frac{1}{1-\alpha} \int_{\text{VaR}_\alpha(X)}^{\infty} I(x)dF_X(x) + (1+\theta)\,\mathbb{E}[I(X)]. \tag{9}$$

Therefore, we have from (6) and (9) that

$$
\begin{aligned}
\text{LVaR}_\alpha(T_I(X)) \ =\ & \omega \cdot \text{TVaR}_\alpha(T_I(X)) + (1-\omega) \cdot \text{VaR}_\alpha(T_I(X)) \\
=\ & \omega\Psi(\alpha) + \text{VaR}_\alpha(X) - (1-\omega)I(\text{VaR}_\alpha(X)) \\
& + (1+\theta)\,\mathbb{E}[I(X)] - \frac{\omega}{1-\alpha} \int_{\text{VaR}_\alpha(X)}^{\infty} I(x)dF_X(x), \tag{10}
\end{aligned}
$$

or equivalently,

$$
\begin{aligned}
\text{LVaR}_\alpha(T_I(X)) \ =\ & \omega\Psi(\alpha) + \text{VaR}_\alpha(X) - (1-\omega)I(\text{VaR}_\alpha(X)) \\
& + (1+\theta)\int_{0}^{\text{VaR}_\alpha(X)} I(x)dF_X(x) + \delta \int_{\text{VaR}_\alpha(X)}^{\infty} I(x)dF_X(x), \tag{11}
\end{aligned}
$$

where $\Psi(\alpha) \overset{\Delta}{=} \frac{1}{1-\alpha} \mathbb{E}\big[(X - \text{VaR}_\alpha(X))_+\big]$.

To proceed, define

$$\mathscr{D}_1 \overset{\Delta}{=} \{(a,b) \mid 0 \leq a \leq \text{VaR}_\alpha(X) \leq b, \quad b - a \leq L\}, \tag{12}$$

$$\mathscr{G}_1 \overset{\Delta}{=} \{g(x;a,b) \mid g(x;a,b) = (x-a)_+ - (x-b)_+, \quad x \geq 0, \quad (a,b) \in \mathscr{D}_1\}. \tag{13}$$

We can easily verify that $\mathscr{G}_1 \subset \mathscr{F}_1$.

**Lemma 2.** *For any confidence level $\alpha \in (p_0, 1)$ and $I(x) \in \mathscr{F}_1$, there always exists a function $g(x;a,b) \in \mathscr{G}_1$ such that $\text{LVaR}_\alpha(T_g(X)) \leq \text{LVaR}_\alpha(T_I(X))$.*

**Remark 3.** *By Lemma 2, (4) reduces to the following optimal problem:*

$$\text{LVaR}_\alpha(T_{g^*}(X)) = \min_{g \in \mathscr{G}_1}\{\text{LVaR}_\alpha(T_g(X))\}. \tag{14}$$

**Theorem 1.** *For any confidence level $\alpha \in (p_0, 1)$, the following results hold according to the sign of $\delta$.*

(i) *For the case of $\delta < 0$, let $a_0$ be the solution of the equation*

$$1 - (1 + \theta)S_X(a) + \delta S_X(a + L) = 0$$

*with respect to a.*

(a) *If $\mathrm{VaR}_\alpha(X) - L \leq a_0$, the minimum of $\mathrm{LVaR}_\alpha(T_I(X))$ is attained at $I^*(x) = (x - a_0)_+ - (x - (a_0 + L))_+$ and*

$$\min_{I \in \mathscr{F}_1}\{\mathrm{LVaR}_\alpha(T_I(X))\} = \omega\Psi(\alpha) + a_0 + (1 + \theta)\int_{a_0}^{\mathrm{VaR}_\alpha(X)} S_X(x)dx + \delta\int_{\mathrm{VaR}_\alpha(X)}^{a_0 + L} S_X(x)dx;$$

(b) *If $\mathrm{VaR}_\alpha(X) - L > a_0$, the minimum of $\mathrm{LVaR}_\alpha(T_I(X))$ is attained at $I^*(x) = (x - (\mathrm{VaR}_\alpha(X) - L))_+ - (x - \mathrm{VaR}_\alpha(X))_+$ and*

$$\min_{I \in \mathscr{F}_1}\{\mathrm{LVaR}_\alpha(T_I(X))\} = \omega\Psi(\alpha) + \mathrm{VaR}_\alpha(X) - L + (1 + \theta)\int_{\mathrm{VaR}_\alpha(X) - L}^{\mathrm{VaR}_\alpha(X)} S_X(x)dx.$$

(ii) *For the case of $\delta = 0$,*

(a) *If $\mathrm{VaR}_\alpha(X) - L \leq \mathrm{VaR}_{\theta*}(X)$, the minimum of $\mathrm{LVaR}_\alpha(T_I(X))$ is attained at $I^*(x) = (x - \mathrm{VaR}_{\theta*}(X))_+ - (x - r)_+$ with $r \in [\mathrm{VaR}_\alpha(X), \mathrm{VaR}_{\theta*}(X) + L]$ and*

$$\min_{I \in \mathscr{F}_1}\{\mathrm{LVaR}_\alpha(T_I(X))\} = \omega\Psi(\alpha) + \mathrm{VaR}_{\theta*}(X) + (1 + \theta)\int_{\mathrm{VaR}_{\theta*}(X)}^{\mathrm{VaR}_\alpha(X)} S_X(x)dx;$$

(b) *If $\mathrm{VaR}_\alpha(X) - L > \mathrm{VaR}_{\theta*}(X)$, the minimum of $\mathrm{LVaR}_\alpha(T_I(X))$ is attained at $I^*(x) = (x - (\mathrm{VaR}_\alpha(X) - L))_+ - (x - \mathrm{VaR}_\alpha(X))_+$ and*

$$\min_{I \in \mathscr{F}_1}\{\mathrm{LVaR}_\alpha(T_I(X))\} = \omega\Psi(\alpha) + \mathrm{VaR}_\alpha(X) - L + (1 + \theta)\int_{\mathrm{VaR}_\alpha(X) - L}^{\mathrm{VaR}_\alpha(X)} S_X(x)dx.$$

(iii) *For the case of $\delta > 0$,*

(a) *If $\alpha \leq \theta^*$, the minimum of $\mathrm{LVaR}_\alpha(T_I(X))$ is attained at $I^*(x) = 0$ and*

$$\min_{I \in \mathscr{F}_1}\{\mathrm{LVaR}_\alpha(T_I(X))\} = \omega\Psi(\alpha) + \mathrm{VaR}_\alpha(X);$$

(b) *If $\alpha > \theta^*$ and $\mathrm{VaR}_\alpha(X) - L \leq \mathrm{VaR}_{\theta*}(X)$, the minimum of $\mathrm{LVaR}_\alpha(T_I(X))$ is attained at $I^*(x) = (x - \mathrm{VaR}_{\theta*}(X))_+ - (x - \mathrm{VaR}_\alpha(X))_+$ and*

$$\min_{I \in \mathscr{F}_1}\{\mathrm{LVaR}_\alpha(T_I(X))\} = \omega\Psi(\alpha) + \mathrm{VaR}_{\theta*}(X) + (1 + \theta)\int_{\mathrm{VaR}_{\theta*}(X)}^{\mathrm{VaR}_\alpha(X)} S_X(x)dx;$$

(c) *If $\alpha > \theta^*$ and $\mathrm{VaR}_\alpha(X) - L > \mathrm{VaR}_{\theta*}(X)$, the minimum of $\mathrm{LVaR}_\alpha(T_I(X))$ is attained at $I^*(x) = (x - (\mathrm{VaR}_\alpha(X) - L))_+ - (x - \mathrm{VaR}_\alpha(X))_+$ and*

$$\min_{I \in \mathscr{F}_1}\{\mathrm{LVaR}_\alpha(T_I(X))\} = \omega\Psi(\alpha) + \mathrm{VaR}_\alpha(X) - L + (1 + \theta)\int_{\mathrm{VaR}_\alpha(X) - L}^{\mathrm{VaR}_\alpha(X)} S_X(x)dx.$$

**Remark 4.** *For $\delta > 0$, the optimal ceded loss function of the three cases can be rewritten in an unified form*

$$I^*(x) = (x - r)_+ - (x - \mathrm{VaR}_\alpha(X))_+,$$

*where $r = \max\{\mathrm{VaR}_\alpha(X) - L, \min\{\mathrm{VaR}_\alpha(X), \mathrm{VaR}_{\theta*}(X)\}\}$. If the safety loading factor $\theta$ is large enough, the optimal ceded loss function is $I^*(x) = 0$, which means that the insurer purchases no reinsurance.*

*3.2. Optimal Reinsurance under the Constraint of $\mathscr{F}_2$*

In this section, we solve (3) under the constraint of $\mathscr{F}_2$:

$$\text{LVaR}_\alpha(T_{I^*}(X)) = \min_{I \in \mathscr{F}_2} \{\text{LVaR}_\alpha(T_I(X))\}. \tag{15}$$

First, we introduce the following notations:

$$\psi(a,b) \stackrel{\Delta}{=} b - a - (1+\theta) \int_a^b S_X(x)dx, \tag{16}$$

$$\mathscr{D}_2 \stackrel{\Delta}{=} \{(a,b) \mid 0 \le a \le \text{VaR}_\alpha(X), b \ge a, \psi(a,b) \le K\}, \tag{17}$$

$$\mathscr{G}_2 \stackrel{\Delta}{=} \{g(x;a,b) \mid g(x;a,b) = (x-a)_+ - (x-b)_+, \ x \ge 0, \ (a,b) \in \mathscr{D}_2\}. \tag{18}$$

We can easily verify that $\mathscr{G}_2 \subset \mathscr{F}_2$.

Before solving (15), we discuss properties of $\psi(a,b)$ needed for future analysis. For any $a_1 \in \mathbb{R}$, taking the partial derivative of $\psi(a_1,b)$ with respect to $b$, we have

$$\frac{\partial \psi(a_1,b)}{\partial b} = 1 - (1+\theta)S_X(b)$$

so that

$$b \gtreqless \text{VaR}_{\theta^*}(X) \Leftrightarrow \frac{\partial \psi(a_1,b)}{\partial b} \gtreqless 0. \tag{19}$$

From (19), we see that $\phi(a_1,b)$ is convex with respect to $b$.

For $g(X;a_1,b) \in \mathscr{G}_2$, we have $g(X;a_1,b) \le X$, which implies that $\mathbb{E}\,g(X;a_1,b) \le \mathbb{E}[X]$. Noting that $\mathbb{E}[g(X;a_1,b)] = \int_{a_1}^b S_X(x)dx$, we have from (17) that

$$\psi(a_1,b) \ge b - a_1 - (1+\theta)\,\mathbb{E}[X].$$

Moreover, as $b \to \infty$, we have

$$b - a_1 - (1+\theta)\,\mathbb{E}[X] \to \infty$$

and

$$\psi(a_1,b) \to \infty.$$

It is obvious that $\psi(a_1,a_1) = 0$ by the definition of $\psi(a,b)$. Thus, by the continuity and convexity of $\psi(a_1,b)$ with respect to $b$, there exists a unique $b_1 > \max\{a_1, \text{VaR}_{\theta^*}(X)\}$ such that $\psi(a_1,b_1) = K$ and $\psi(a_1,b) \le K$ for $b \in [a_1,b_1]$.

For any $a \in \mathbb{R}$, let $\beta(a)$ be the unique solution to the equation $\psi(a,b) = K$ with $b \in (\max\{a, \text{VaR}_{\theta^*}(X)\}, \infty)$. For $b > a$, taking the derivatives with respect to $a$ on both sides of the equation $\psi(a,b) = K$ entails

$$b' = \beta'(a) = \frac{1 - (1+\theta)S_X(a)}{1 - (1+\theta)S_X(b)}.$$

Furthermore, we can verify that $1 - (1+\theta)S_X(b) > 0$ when $b = \beta(a) > \text{VaR}_{\theta^*}(X)$. Thus,

$$a \lesseqgtr \text{VaR}_{\theta^*}(X) \leftrightarrow \beta'(a) \lesseqgtr 0,$$

which implies that $\beta(a)$ is convex with respect to $a$. Therefore, the set $\mathscr{D}_2$ can be rewritten as

$$\mathscr{D}_0 \stackrel{\Delta}{=} \{(a,b) \mid 0 \le a \le \text{VaR}_\alpha(X), a \le b \le \beta(a)\}. \tag{20}$$

**Lemma 3.** *For any confidence level $\alpha \in (p_0, 1)$ and $I(x) \in \mathscr{F}_2$, there always exists a function $g(x; a, b) \in \mathscr{G}_2$ such that $\mathrm{LVaR}_\alpha\left(T_g(X)\right) \leq \mathrm{LVaR}_\alpha\left(T_I(X)\right)$.*

**Remark 5.** *By Lemma 3, we can solve (15) by considering the following optimal problem:*

$$\mathrm{LVaR}_\alpha\left(T_{g^*}(X)\right) = \min_{g \in \mathscr{G}_2}\left\{\mathrm{LVaR}_\alpha\left(T_g(X)\right)\right\}. \tag{21}$$

To facilitate subsequent analysis, let $b_0^* = \beta(\mathrm{VaR}_{\theta*}(X))$, that is, let $b_0^*$ be the unique solution to

$$b - \mathrm{VaR}_{\theta*}(X) - (1 + \theta)\int_{\mathrm{VaR}_{\theta*}(X)}^b S_X(x)dx = K$$

subjected to $b > \mathrm{VaR}_{\theta*}(X)$. Differentiating both sides with respect to $K$ yields

$$\frac{\partial b}{\partial K} = \frac{1}{1 - (1 + \theta)S_X(b)} > 0,$$

which implies that $b$ is increasing with respect to $K$. If $b_0^* < \mathrm{VaR}_\alpha(X)$, the equation $\beta(a) = \mathrm{VaR}_\alpha(X)$ must have two solutions in $(-\infty, \mathrm{VaR}_\alpha(X)]$, denoted by $a_0^*$ and $a_1^*$ with $a_0^* < a_1^*$. Furthermore, let $b_1^* = \beta(\mathrm{VaR}_\alpha(X))$, then $b_1^* > \mathrm{VaR}_\alpha(X)$.

**Theorem 2.** *For any confidence level $\alpha \in (p_0, 1)$, the optimal decision in (15) for the insurer is given as follows according to the sign of $\delta$.*

(i) *For the case of $\delta < 0$,*

 (a) *If $b_0^* \geq \mathrm{VaR}_\alpha(X)$, the minimum of $\mathrm{LVaR}_\alpha\left(T_I(X)\right)$ is attained at $I^*(x) = (x - \mathrm{VaR}_{\theta*}(X))_+ - (x - b_0^*)_+$ and*

$$\min_{I \in \mathscr{F}_2}\left\{\mathrm{LVaR}_\alpha\left(T_I(X)\right)\right\} = \omega\Psi(\alpha) - K + h(b_0^*),$$

 *where $h(b) \triangleq b - \frac{\omega}{1 - \alpha}\int_{\mathrm{VaR}_\alpha(X)}^b S_X(x)dx$;*

 (b) *If $b_0^* < \mathrm{VaR}_\alpha(X)$, the minimum of $\mathrm{LVaR}_\alpha\left(T_I(X)\right)$ is attained at $I^*(x) = (x - r)_+ - (x - \beta(r))_+$ with $r \in \left[\max\{0, a_0^*\}, a_1^*\right]$, and*

$$\min_{I \in \mathscr{F}_2}\left\{\mathrm{LVaR}_\alpha\left(T_I(X)\right)\right\} = \omega\Psi(\alpha) + \mathrm{VaR}_\alpha(X) - K.$$

(ii) *For the case of $\delta = 0$,*

 (a) *If $\alpha = \theta^*$, the minimum of $\mathrm{LVaR}_\alpha\left(T_I(X)\right)$ is attained at $I^*(x) = (x - \mathrm{VaR}_\alpha(X))_+ - (x - r)_+$ for any $r \in \left[\mathrm{VaR}_\alpha(X), b_1^*\right]$, and*

$$\min_{I \in \mathscr{F}_2}\left\{\mathrm{LVaR}_\alpha\left(T_I(X)\right)\right\} = \omega\Psi(\alpha) + \mathrm{VaR}_\alpha(X);$$

 (b) *If $\alpha > \theta^*$ and $b_0^* \geq \mathrm{VaR}_\alpha(X)$, the minimum of $\mathrm{LVaR}_\alpha\left(T_I(X)\right)$ is attained at $I^*(x) = (x - \mathrm{VaR}_{\theta*}(X))_+ - (x - r)_+$ for any $r \in [\mathrm{VaR}_{\theta*}(X), b_0^*]$, and*

$$\min_{I \in \mathscr{F}_2}\left\{\mathrm{LVaR}_\alpha\left(T_I(X)\right)\right\} = \omega\Psi(\alpha) + \mathrm{VaR}_{\theta*}(X) + (1 + \theta)\int_{\mathrm{VaR}_{\theta*}(X)}^{\mathrm{VaR}_\alpha(X)} S_X(x)dx;$$

 (c) *If $\alpha > \theta^*$ and $b_0^* < \mathrm{VaR}_\alpha(X)$, the minimum of $\mathrm{LVaR}_\alpha\left(T_I(X)\right)$ is attained at $I^*(x) = (x - r)_+ - (x - \beta(r))_+$ with $r \in \left[\max\{0, a_0^*\}, a_1^*\right]$, and*

$$\min_{I \in \mathscr{F}_2}\left\{\mathrm{LVaR}_\alpha\left(T_I(X)\right)\right\} = \omega\Psi(\alpha) + \mathrm{VaR}_\alpha(X) - K.$$

(iii) *For the case of $\delta > 0$,*

(a) *If $\alpha \leq \theta^*$, the minimum of $\mathrm{LVaR}_\alpha(T_I(X))$ is attained at $I^*(x) = 0$ and*

$$\min_{I \in \mathscr{F}_2}\{\mathrm{LVaR}_\alpha(T_I(X))\} = \omega\Psi(\alpha) + \mathrm{VaR}_\alpha(X);$$

(b) *If $\alpha > \theta^*$ and $b_0^* \geq \mathrm{VaR}_\alpha(X)$, the minimum of $\mathrm{LVaR}_\alpha(T_I(X))$ is attained at $I^*(x) = (x - \mathrm{VaR}_{\theta^*}(X))_+ - (x - \mathrm{VaR}_\alpha(X))_+$, and*

$$\min_{I \in \mathscr{F}_2}\{\mathrm{LVaR}_\alpha(T_I(X))\} = \omega\Psi(\alpha) + \mathrm{VaR}_{\theta^*}(X) + (1 + \theta)\int_{\mathrm{VaR}_{\theta^*}(X)}^{\mathrm{VaR}_\alpha(X)} S_X(x)dx;$$

(c) *If $\alpha > \theta^*$ and $b_0^* < \mathrm{VaR}_\alpha(X)$, the minimum of $\mathrm{LVaR}_\alpha(T_I(X))$ is attained at $I^*(x) = (x - r)_+ - (x - \beta(r))_+$ with $r \in \left[\max\{0, a_0^*\}, a_1^*\right]$, and*

$$\min_{I \in \mathscr{F}_2}\{\mathrm{LVaR}_\alpha(T_I(X))\} = \omega\Psi(\alpha) + \mathrm{VaR}_\alpha(X) - K.$$

**Remark 6.** *Theorem 2 shows that for $\delta > 0$ larger safety loading factor $\theta$ leads to no reinsurance. Otherwise, the two layer reinsurance is optimal.*

## 4. Numerical Illustrations

In this section, we present some examples to illustrate the results relying on Theorems 1 and 2. We also compare them with the results in Lu et al. (2016) by setting different confidence levels and different weight coefficients.

In the first two examples, we consider that $X$ follows from an exponential distribution or a normal distribution. Both examples show that under the optimal reinsurance policy larger weight coefficients lead to less ceded losses (or fewer reinsurance premiums), and, hence, the total risks measured by the LVaRs lie somewhere between that measured by the VaR and the TVaR. As a new family of risk measures, LVaR provides flexible risk assessments. For example, the conservative insurer may prefer the larger weight coefficient in practice. Additionally, most risk managers would argue that the heavy-tailed distribution is riskier than the light-tailed distribution since larger losses are more likely to occur when the distribution is heavy-tailed. Therefore, we consider cases of heavy-tailed distributions presented in Examples 3–5. From tables given later, we can conclude that the absolute values of the difference between $\mathrm{VaR}_\alpha(T_{I^*}(X))$ and $\mathrm{TVaR}_\alpha(T_{I^*}(X))$ for heavy-tailed distributions are larger than those for light-tailed distributions. Moreover, the higher confidence level leads to the larger absolute value of the difference. In such cases, the new family of flexible risk measures LVaR permits risk managers to seek an equilibrium between different demands better. Also, we can see that for cases of heavy-tailed distributions the optimal ceded losses are larger, in other words, the insurer needs to hold less losses and pay more to the reinsurer for larger ceded losses.

**Example 1.** *We assume that $X$ is an exponential random variable with survival function $S_X(x) = e^{-0.01x}$. The optimal reinsurance strategy under risk measures with both types of constraints are given in Tables 1 and 2 by setting $\theta = 3$, $L = 120$, and $K = 160$, where the risk measure $\rho$ represents VaR, LVaR, and TVaR. $\rho_\alpha(T_{I^*}(X))$ denotes the risk under the optimal reinsurance strategy $I^*(x)$ and given confidence level $\alpha$.*

**Example 2.** *Suppose that $X \sim \mathbb{N}(40, 100)$. The optimal reinsurance solutions under risk measures with both types of constraints are given in Tables 3 and 4 by setting $\theta = 4$, $L = 100$, and $K = 120$, where the risk measure $\rho$ represents VaR, LVaR, and TVaR. $\rho_\alpha(T_{I^*}(X))$ denotes the risk under the optimal reinsurance strategy $I^*(x)$ and given confidence level $\alpha$.*

**Example 3.** *Suppose that X follows Pareto distribution with $F(x) = 1 - \left(\frac{x}{\sigma}\right)^{-1/\gamma}$, $0 < \sigma \leq x$. For $\gamma = 1/3$, $\sigma = 120$, the optimal reinsurance solutions under risk measures with both types of constraints are given in Tables 5 and 6 by setting $\theta = 4$, $L = 100$, and $K = 140$, where the*

risk measure $\rho$ represents VaR, LVaR, and TVaR. $\rho_\alpha(T_{I^*}(X))$ denotes the risk under the optimal reinsurance strategy $I^*(x)$ and given confidence level $\alpha$.

**Example 4.** *We suppose that X follows Fréchet distribution* $F(x) = \exp\left\{-\left(\frac{x-\mu}{\sigma}\right)^{-1/\gamma}\right\}$ *with* $\mu = 5$, $\gamma = 1/3$, *and* $\sigma = 50$. *The optimal reinsurance solutions under risk measures with both types of constraints are given in Tables 7 and 8 by setting* $\theta = 3$, $L = 100$, *and* $K = 120$, *where the risk measure* $\rho$ *represents VaR, LVaR, and TVaR.* $\rho_\alpha(T_{I^*}(X))$ *denotes the risk under the optimal reinsurance strategy* $I^*(x)$ *and given confidence level* $\alpha$.

**Example 5.** *We suppose that X has Burr distribution* $F(x) = 1 - \left(\frac{\eta + x^{-\rho\alpha}}{\eta}\right)^{1/\rho}$ *with* $\alpha = 1$, $\rho = 3$, *and* $\eta = 40$. *The optimal reinsurance solutions under risk measures with both types of constraints are given in Tables 9 and 10 by setting* $\theta = 4$, $L = 120$ *and* $K = 160$, *where the risk measure* $\rho$ *represents VaR, LVaR, and TVaR.* $\rho_\alpha(T_{I^*}(X))$ *denotes the risk under the optimal reinsurance strategy* $I^*(x)$ *and given confidence level* $\alpha$.

**Table 1.** Optimal solutions under different risk measures with the constraint of $\mathscr{F}_1$ under the settings that $X \sim F(x) = 1 - e^{-0.01x}$, $x > 0$, $\theta = 3$, and $L = 120$.

| $\alpha$ | The Risk Measure $\rho$ | $\rho_\alpha(T_{I^*}(X))$ | $I^*(x)$ |
|---|---|---|---|
| 0.90 | VaR | 198.629 | $(x - 138.629)_+ - (x - 230.259)_+$ |
| | LVaR($\omega = 0.2$) | 218.629 | $(x - 138.629)_+ - (x - 230.259)_+$ |
| | LVaR($\omega = 0.5$) | 245.889 | $(x - 145.889)_+ - (x - 265.889)_+$ |
| | LVaR($\omega = 0.8$) | 264.958 | $(x - 164.958)_+ - (x - 284.958)_+$ |
| | TVaR | 275.909 | $(x - 175.909)_+ - (x - 295.909)_+$ |
| 0.95 | VaR | 225.976 | $(x - 179.573)_+ - (x - 299.573)_+$ |
| | LVaR($\omega = 0.2$) | 245.976 | $(x - 179.573)_+ - (x - 299.573)_+$ |
| | LVaR($\omega = 0.5$) | 275.976 | $(x - 179.573)_+ - (x - 299.573)_+$ |
| | LVaR($\omega = 0.8$) | 303.003 | $(x - 203.003)_+ - (x - 323.003)_+$ |
| | TVaR | 317.692 | $(x - 217.692)_+ - (x - 337.692)_+$ |
| 0.97 | VaR | 258.497 | $(x - 230.656)_+ - (x - 350.656)_+$ |
| | LVaR($\omega = 0.2$) | 278.497 | $(x - 230.656)_+ - (x - 350.656)_+$ |
| | LVaR($\omega = 0.5$) | 308.497 | $(x - 230.656)_+ - (x - 350.656)_+$ |
| | LVaR($\omega = 0.8$) | 338.205 | $(x - 238.205)_+ - (x - 358.205)_+$ |
| | TVaR | 355.218 | $(x - 255.218)_+ - (x - 375.218)_+$ |
| 0.99 | VaR | 349.798 | $(x - 340.517)_+ - (x - 460.517)_+$ |
| | LVaR($\omega = 0.2$) | 369.798 | $(x - 340.517)_+ - (x - 460.517)_+$ |
| | LVaR($\omega = 0.5$) | 399.798 | $(x - 340.517)_+ - (x - 460.517)_+$ |
| | LVaR($\omega = 0.8$) | 429.798 | $(x - 340.517)_+ - (x - 460.517)_+$ |
| | TVaR | 449.392 | $(x - 349.392)_+ - (x - 469.392)_+$ |
| 0.999 | VaR | 571.704 | $(x - 570.776)_+ - (x - 690.776)_+$ |
| | LVaR($\omega = 0.2$) | 591.703 | $(x - 570.776)_+ - (x - 690.776)_+$ |
| | LVaR($\omega = 0.5$) | 621.703 | $(x - 570.776)_+ - (x - 690.776)_+$ |
| | LVaR($\omega = 0.8$) | 651.703 | $(x - 570.776)_+ - (x - 690.776)_+$ |
| | TVaR | 671.698 | $(x - 571.699)_+ - (x - 691.699)_+$ |

**Table 2.** Optimal solutions under different risk measures with the constraint of $\mathscr{F}_2$ under the settings that $X \sim F(x) = 1 - e^{-0.01x}$, $x > 0$, $\theta = 3$, and $K = 160$.

| $\alpha$ | The Risk Measure $\rho$ | $\rho_\alpha(T_{I^*}(X))$ | $I^*(x)$ |
|---|---|---|---|
| 0.90 | VaR | 198.629 | $(x - 138.629)_+ - (x - 230.259)_+$ |
| | LVaR($\omega = 0.2$) | 218.629 | $(x - 138.629)_+ - (x - 230.259)_+$ |
| | LVaR($\omega = 0.5$) | 240.642 | $(x - 138.629)_+ - (x - 390.580)_+$ |
| | LVaR($\omega = 0.8$) | 246.679 | $(x - 138.629)_+ - (x - 390.580)_+$ |
| | TVaR | 250.704 | $(x - 138.629)_+ - (x - 390.580)_+$ |

**Table 2.** *Cont.*

| $\alpha$ | The Risk Measure $\rho$ | $\rho_\alpha(T_{I^*}(X))$ | $I^*(x)$ |
|---|---|---|---|
| 0.95 | VaR | 218.629 | $(x-138.629)_+ - (x-299.573)_+$ |
| | LVaR($\omega=0.2$) | 238.629 | $(x-138.629)_+ - (x-r)_+$ with $r \in [138.629, 390.580]$ |
| | LVaR($\omega=0.5$) | 250.704 | $(x-138.629)_+ - (x-390.580)_+$ |
| | LVaR($\omega=0.8$) | 262.779 | $(x-138.629)_+ - (x-390.580)_+$ |
| | TVaR | 270.829 | $(x-138.629)_+ - (x-390.580)_+$ |
| 0.97 | VaR | 226.629 | $(x-138.629)_+ - (x-350.656)_+$ |
| | LVaR($\omega=0.2$) | 243.996 | $(x-138.629)_+ - (x-390.580)_+$ |
| | LVaR($\omega=0.5$) | 264.121 | $(x-138.629)_+ - (x-390.580)_+$ |
| | LVaR($\omega=0.8$) | 284.246 | $(x-138.629)_+ - (x-390.580)_+$ |
| | TVaR | 297.663 | $(x-138.629)_+ - (x-390.580)_+$ |
| 0.99 | VaR | 300.517 | $(x-r)_+ - (x-\beta(r))_+$ with $r \in [42.186, 280.255]$ and $\beta(r) \in [390.580, 460.517]$ |
| | LVaR($\omega=0.2$) | 320.517 | $(x-r)_+ - (x-\beta(r))_+$ with $r \in [42.186, 280.255]$ and $\beta(r) \in [390.580, 460.517]$ |
| | LVaR($\omega=0.5$) | 350.517 | $(x-r)_+ - (x-\beta(r))_+$ with $r \in [42.186, 280.255]$ and $\beta(r) \in [390.580, 460.517]$ |
| | LVaR($\omega=0.8$) | 380.517 | $(x-r)_+ - (x-\beta(r))_+$ with $r \in [42.186, 280.255]$ and $\beta(r) \in [390.580, 460.517]$ |
| | TVaR | 400.517 | $(x-r)_+ - (x-\beta(r))_+$ with $r \in [42.186, 280.255]$ and $\beta(r) \in [390.580, 460.517]$ |
| 0.999 | VaR | 530.776 | $(x-r)_+ - (x-\beta(r))_+$ with $r \in [0, 529.162]$ and $\beta(r) \in [390.580, 690.776]$ |
| | LVaR($\omega=0.2$) | 550.775 | $(x-r)_+ - (x-\beta(r))_+$ with $r \in [0, 529.162]$ and $\beta(r) \in [390.580, 690.776]$ |
| | LVaR($\omega=0.5$) | 580.775 | $(x-r)_+ - (x-\beta(r))_+$ with $r \in [0, 529.162]$ and $\beta(r) \in [390.580, 690.776]$ |
| | LVaR($\omega=0.8$) | 610.775 | $(x-r)_+ - (x-\beta(r))_+$ with $r \in [0, 529.162]$ and $\beta(r) \in [390.580, 690.776]$ |
| | TVaR | 630.774 | $(x-r)_+ - (x-\beta(r))_+$ with $r \in [0, 529.162]$ and $\beta(r) \in [390.580, 690.776]$ |

**Table 3.** Optimal solutions under different risk measures with the constraint of $\mathscr{F}_1$ under the settings that $X \sim \mathbb{N}(40, 100)$, $\theta = 4$, and $L = 100$.

| $\alpha$ | The Risk Measure $\rho$ | $\rho_\alpha(T_{I^*}(X))$ | $I^*(x)$ |
|---|---|---|---|
| 0.90 | VaR | 156.309 | $(x-124.162)_+ - (x-168.155)_+$ |
| | LVaR($\omega=0.2$) | 165.778 | $(x-124.162)_+ - (x-168.155)_+$ |
| | LVaR($\omega=0.5$) | 179.981 | $(x-124.162)_+ - (x-r)_+$ with $r \in [168.155, 224.162]$ |
| | LVaR($\omega=0.8$) | 183.531 | $(x-130.423)_+ - (x-230.423)_+$ |
| | TVaR | 185.618 | $(x-133.936)_+ - (x-233.936)_+$ |
| 0.95 | VaR | 169.535 | $(x-124.162)_+ - (x-204.485)_+$ |
| | LVaR($\omega=0.2$) | 177.892 | $(x-124.162)_+ - (x-204.485)_+$ |
| | LVaR($\omega=0.5$) | 185.618 | $(x-133.937)_+ - (x-233.937)_+$ |
| | LVaR($\omega=0.8$) | 190.944 | $(x-142.453)_+ - (x-242.453)_+$ |
| | TVaR | 193.931 | $(x-146.997)_+ - (x-246.997)_+$ |
| 0.97 | VaR | 174.278 | $(x-128.079)_+ - (x-228.079)_+$ |
| | LVaR($\omega=0.2$) | 182.024 | $(x-128.079)_+ - (x-228.079)_+$ |
| | LVaR($\omega=0.5$) | 191.469 | $(x-143.261)_+ - (x-243.261)_+$ |
| | LVaR($\omega=0.8$) | 198.218 | $(x-153.286)_+ - (x-253.286)_+$ |
| | TVaR | 201.874 | $(x-158.466)_+ - (x-258.466)_+$ |
| 0.99 | VaR | 192.459 | $(x-172.635)_+ - (x-272.635)_+$ |
| | LVaR($\omega=0.2$) | 199.236 | $(x-172.635)_+ - (x-272.635)_+$ |
| | LVaR($\omega=0.5$) | 209.402 | $(x-172.635)_+ - (x-272.635)_+$ |
| | LVaR($\omega=0.8$) | 218.730 | $(x-180.772)_+ - (x-280.772)_+$ |
| | TVaR | 223.571 | $(x-186.826)_+ - (x-286.826)_+$ |
| 0.999 | VaR | 252.207 | $(x-249.023)_+ - (x-349.023)_+$ |
| | LVaR($\omega=0.2$) | 257.744 | $(x-249.023)_+ - (x-349.023)_+$ |
| | LVaR($\omega=0.5$) | 266.050 | $(x-249.023)_+ - (x-349.023)_+$ |
| | LVaR($\omega=0.8$) | 274.356 | $(x-249.023)_+ - (x-349.023)_+$ |
| | TVaR | 279.786 | $(x-251.536)_+ - (x-351.536)_+$ |

**Table 4.** Optimal solutions under different risk measures with the constraint of $\mathscr{F}_2$ under the settings that $X \sim \mathbb{N}(40, 100)$, $\theta = 4$, and $K = 120$.

| $\alpha$ | The Risk Measure $\rho$ | $\rho_\alpha(T_{I^*}(X))$ | $I^*(x)$ |
|---|---|---|---|
| 0.90 | VaR | 156.309 | $(x - 124.162)_+ - (x - 168.155)_+$ |
| | LVaR($\omega = 0.2$) | 165.778 | $(x - 124.162)_+ - (x - 168.155)_+$ |
| | LVaR($\omega = 0.5$) | 179.981 | $(x - 124.162)_+ - (x - r)_+$ with $r \in [124.162, 299.231]$ |
| | LVaR($\omega = 0.8$) | 180.431 | $(x - 124.162)_+ - (x - 299.231)_+$ |
| | TVaR | 180.731 | $(x - 124.162)_+ - (x - 299.231)_+$ |
| 0.95 | VaR | 169.535 | $(x - 124.162)_+ - (x - 204.485)_+$ |
| | LVaR($\omega = 0.2$) | 177.892 | $(x - 124.162)_+ - (x - 204.485)_+$ |
| | LVaR($\omega = 0.5$) | 180.731 | $(x - 124.162)_+ - (x - 299.231)_+$ |
| | LVaR($\omega = 0.8$) | 181.631 | $(x - 124.162)_+ - (x - 299.231)_+$ |
| | TVaR | 182.231 | $(x - 124.162)_+ - (x - 299.231)_+$ |
| 0.97 | VaR | 174.172 | $(x - 124.162)_+ - (x - 228.079)_+$ |
| | LVaR($\omega = 0.2$) | 180.231 | $(x - 124.162)_+ - (x - 299.231)_+$ |
| | LVaR($\omega = 0.5$) | 181.731 | $(x - 124.162)_+ - (x - 299.231)_+$ |
| | LVaR($\omega = 0.8$) | 183.231 | $(x - 124.162)_+ - (x - 299.231)_+$ |
| | TVaR | 184.231 | $(x - 124.162)_+ - (x - 299.231)_+$ |
| 0.99 | VaR | 178.287 | $(x - 124.162)_+ - (x - 272.635)_+$ |
| | LVaR($\omega = 0.2$) | 182.231 | $(x - 124.162)_+ - (x - 299.231)_+$ |
| | LVaR($\omega = 0.5$) | 186.732 | $(x - 124.162)_+ - (x - 299.231)_+$ |
| | LVaR($\omega = 0.8$) | 191.232 | $(x - 124.162)_+ - (x - 299.231)_+$ |
| | TVaR | 194.232 | $(x - 124.162)_+ - (x - 299.231)_+$ |
| 0.999 | VaR | 229.023 | $(x - r)_+ - (x - \beta(r))_+$ with $r \in [47.219, 222.453]$ and $\beta(r) \in [299.231, 349.023]$ |
| | LVaR($\omega = 0.2$) | 234.560 | $(x - r)_+ - (x - \beta(r))_+$ with $r \in [47.219, 222.453]$ and $\beta(r) \in [299.231, 349.023]$ |
| | LVaR($\omega = 0.5$) | 242.866 | $(x - r)_+ - (x - \beta(r))_+$ with $r \in [47.219, 222.453]$ and $\beta(r) \in [299.231, 349.023]$ |
| | LVaR($\omega = 0.8$) | 251.172 | $(x - r)_+ - (x - \beta(r))_+$ with $r \in [47.219, 222.453]$ and $\beta(r) \in [299.231, 349.023]$ |
| | TVaR | 256.709 | $(x - r)_+ - (x - \beta(r))_+$ with $r \in [47.219, 222.453]$ and $\beta(r) \in [299.231, 349.023]$ |

**Table 5.** Optimal solutions under different risk measures with the constraint of $\mathscr{F}_1$ under the settings that $X \sim F(x) = 1 - \left(\frac{x}{120}\right)^{-3}$, $\theta = 4$, and $L = 100$.

| $\alpha$ | The Risk Measure $\rho$ | $\rho_\alpha(T_{I^*}(X))$ | $I^*(x)$ |
|---|---|---|---|
| 0.90 | VaR | 123.163 | $(x - 85.197)_+ - (x - 138.532)_+$ |
| | LVaR($\omega = 0.2$) | 149.016 | $(x - 85.197)_+ - (x - 138.532)_+$ |
| | LVaR($\omega = 0.5$) | 187.796 | $(x - 85.197)_+ - (x - r)_+$ with $r \in [138.532, 235.197]$ |
| | LVaR($\omega = 0.8$) | 207.890 | $(x - 93.182)_+ - (x - 243.182)_+$ |
| | TVaR | 220.799 | $(x - 98.571)_+ - (x - 248.571)_+$ |
| 0.95 | VaR | 147.079 | $(x - 85.197)_+ - (x - 205.730)_+$ |
| | LVaR($\omega = 0.2$) | 179.652 | $(x - 85.197)_+ - (x - 205.730)_+$ |
| | LVaR($\omega = 0.5$) | 220.799 | $(x - 98.571)_+ - (x - 248.571)_+$ |
| | LVaR($\omega = 0.8$) | 257.344 | $(x - 114.895)_+ - (x - 264.895)_+$ |
| | TVaR | 280.031 | $(x - 125.783)_+ - (x - 275.783)_+$ |
| 0.97 | VaR | 164.667 | $(x - 116.196)_+ - (x - 266.196)_+$ |
| | LVaR($\omega = 0.2$) | 203.286 | $(x - 116.196)_+ - (x - 266.196)_+$ |
| | LVaR($\omega = 0.5$) | 261.214 | $(x - 116.196)_+ - (x - 266.196)_+$ |
| | LVaR($\omega = 0.8$) | 315.204 | $(x - 143.696)_+ - (x - 293.696)_+$ |
| | TVaR | 347.493 | $(x - 161.109)_+ - (x - 311.109)_+$ |
| 0.99 | VaR | 299.146 | $(x - 286.991)_+ - (x - 436.991)_+$ |
| | LVaR($\omega = 0.2$) | 354.845 | $(x - 286.991)_+ - (x - 436.991)_+$ |
| | LVaR($\omega = 0.5$) | 438.394 | $(x - 286.991)_+ - (x - 436.991)_+$ |
| | LVaR($\omega = 0.8$) | 521.943 | $(x - 286.991)_+ - (x - 436.991)_+$ |
| | TVaR | 577.121 | $(x - 300.555)_+ - (x - 450.555)_+$ |
| 0.999 | VaR | 930.918 | $(x - 930.000)_+ - (x - 1080.000)_+$ |
| | LVaR($\omega = 0.2$) | 1050.918 | $(x - 930.000)_+ - (x - 1080.000)_+$ |
| | LVaR($\omega = 0.5$) | 1230.918 | $(x - 930.000)_+ - (x - 1080.000)_+$ |
| | LVaR($\omega = 0.8$) | 1410.918 | $(x - 930.000)_+ - (x - 1080.000)_+$ |
| | TVaR | 1530.917 | $(x - 930.984)_+ - (x - 1080.983)_+$ |

**Table 6.** Optimal solutions under different risk measures with the constraint of $\mathscr{F}_2$ under the settings that $X \sim F(x) = 1 - \left(\frac{x}{120}\right)^{-3}$, $\theta = 4$, and $K = 140$.

| $\alpha$ | The Risk Measure $\rho$ | $\rho_\alpha(T_{I^*}(X))$ | $I^*(x)$ |
|---|---|---|---|
| 0.90 | VaR | 123.163 | $(x - 85.197)_+ - (x - 138.532)_+$ |
| | LVaR($\omega = 0.2$) | 149.016 | $(x - 85.197)_+ - (x - 138.532)_+$ |
| | LVaR($\omega = 0.5$) | 187.796 | $(x - 85.197)_+ - (x - r)_+$ with $r \in [85.197, 369.788]$ |
| | LVaR($\omega = 0.8$) | 198.601 | $(x - 85.197)_+ - (x - 369.788)_+$ |
| | TVaR | 205.804 | $(x - 85.197)_+ - (x - 369.788)_+$ |
| 0.95 | VaR | 147.079 | $(x - 85.197)_+ - (x - 205.730)_+$ |
| | LVaR($\omega = 0.2$) | 179.652 | $(x - 85.197)_+ - (x - 205.730)_+$ |
| | LVaR($\omega = 0.5$) | 205.831 | $(x - 85.197)_+ - (x - 369.788)_+$ |
| | LVaR($\omega = 0.8$) | 227.414 | $(x - 85.197)_+ - (x - 369.788)_+$ |
| | TVaR | 241.820 | $(x - 85.197)_+ - (x - 369.788)_+$ |
| 0.97 | VaR | 158.831 | $(x - 85.197)_+ - (x - 266.196)_+$ |
| | LVaR($\omega = 0.2$) | 193.798 | $(x - 85.197)_+ - (x - 369.788)_+$ |
| | LVaR($\omega = 0.5$) | 229.815 | $(x - 85.197)_+ - (x - 369.788)_+$ |
| | LVaR($\omega = 0.8$) | 265.831 | $(x - 85.197)_+ - (x - 369.788)_+$ |
| | TVaR | 289.842 | $(x - 85.197)_+ - (x - 369.788)_+$ |
| 0.99 | VaR | 236.991 | $(x - r)_+ - (x - \beta(r))_+$ with $r \in [15.450, 211.637]$ and $\beta(r) \in [369.788, 436.991]$ |
| | LVaR($\omega = 0.2$) | 292.690 | $(x - r)_+ - (x - \beta(r))_+$ with $r \in [15.450, 211.637]$ and $\beta(r) \in [369.788, 436.991]$ |
| | LVaR($\omega = 0.5$) | 376.238 | $(x - r)_+ - (x - \beta(r))_+$ with $r \in [15.450, 211.637]$ and $\beta(r) \in [369.788, 436.991]$ |
| | LVaR($\omega = 0.8$) | 459.787 | $(x - r)_+ - (x - \beta(r))_+$ with $r \in [15.450, 211.637]$ and $\beta(r) \in [369.788, 436.991]$ |
| | TVaR | 515.486 | $(x - r)_+ - (x - \beta(r))_+$ with $r \in [15.450, 211.637]$ and $\beta(r) \in [369.788, 436.991]$ |
| 0.999 | VaR | 880.000 | $(x - r)_+ - (x - \beta(r))_+$ with $r \in [0, 878.669]$ and $\beta(r) \in [369.788, 1080]$ |
| | LVaR($\omega = 0.2$) | 1000.000 | $(x - r)_+ - (x - \beta(r))_+$ with $r \in [0, 878.669]$ and $\beta(r) \in [369.788, 1080]$ |
| | LVaR($\omega = 0.5$) | 1180.000 | $(x - r)_+ - (x - \beta(r))_+$ with $r \in [0, 878.669]$ and $\beta(r) \in [369.788, 1080]$ |
| | LVaR($\omega = 0.8$) | 1360.000 | $(x - r)_+ - (x - \beta(r))_+$ with $r \in [0, 878.669]$ and $\beta(r) \in [369.788, 1080]$ |
| | TVaR | 1480.000 | $(x - r)_+ - (x - \beta(r))_+$ with $r \in [0, 878.669]$ and $\beta(r) \in [369.788, 1080]$ |

**Table 7.** Optimal solutions under different risk measures with the constraint of $\mathscr{F}_1$ under the settings that $X \sim F(x) = \exp\left\{-\left(\frac{x-5}{50}\right)^{-3}\right\}$, $\theta = 3$, and $L = 100$.

| $\alpha$ | The Risk Measure $\rho$ | $\rho_\alpha(T_{I^*}(X))$ | $I^*(x)$ |
|---|---|---|---|
| 0.90 | VaR | 100.108 | $(x - 80.741)_+ - (x - 110.863)_+$ |
| | LVaR($\omega = 0.2$) | 111.032 | $(x - 80.741)_+ - (x - 110.863)_+$ |
| | LVaR($\omega = 0.5$) | 123.962 | $(x - 81.409)_+ - (x - 181.409)_+$ |
| | LVaR($\omega = 0.8$) | 129.893 | $(x - 83.432)_+ - (x - 183.432)_+$ |
| | TVaR | 133.772 | $(x - 84.797)_+ - (x - 184.797)_+$ |
| 0.95 | VaR | 108.290 | $(x - 80.741)_+ - (x - 139.571)_+$ |
| | LVaR($\omega = 0.2$) | 121.955 | $(x - 80.741)_+ - (x - r)_+$ with $r \in [139.571, 180.741]$ |
| | LVaR($\omega = 0.5$) | 133.772 | $(x - 84.797)_+ - (x - 184.797)_+$ |
| | LVaR($\omega = 0.8$) | 145.064 | $(x - 88.963)_+ - (x - 188.963)_+$ |
| | TVaR | 152.311 | $(x - 91.794)_+ - (x - 191.794)_+$ |
| 0.97 | VaR | 112.261 | $(x - 80.741)_+ - (x - 165.102)_+$ |
| | LVaR($\omega = 0.2$) | 127.273 | $(x - 82.529)_+ - (x - 182.529)_+$ |
| | LVaR($\omega = 0.5$) | 146.287 | $(x - 89.432)_+ - (x - 189.432)_+$ |
| | LVaR($\omega = 0.8$) | 163.912 | $(x - 96.586)_+ - (x - 196.586)_+$ |
| | TVaR | 174.945 | $(x - 101.444)_+ - (x - 201.444)_+$ |
| 0.99 | VaR | 146.303 | $(x - 136.691)_+ - (x - 236.691)_+$ |
| | LVaR($\omega = 0.2$) | 169.542 | $(x - 136.691)_+ - (x - 236.691)_+$ |
| | LVaR($\omega = 0.5$) | 204.401 | $(x - 136.691)_+ - (x - 236.691)_+$ |
| | LVaR($\omega = 0.8$) | 239.259 | $(x - 136.691)_+ - (x - 236.691)_+$ |
| | TVaR | 261.556 | $(x - 148.004)_+ - (x - 248.004)_+$ |
| 0.999 | VaR | 405.479 | $(x - 404.917)_+ - (x - 504.917)_+$ |
| | LVaR($\omega = 0.2$) | 455.486 | $(x - 404.917)_+ - (x - 504.917)_+$ |
| | LVaR($\omega = 0.5$) | 530.496 | $(x - 404.917)_+ - (x - 504.917)_+$ |
| | LVaR($\omega = 0.8$) | 605.506 | $(x - 404.917)_+ - (x - 504.917)_+$ |
| | TVaR | 655.511 | $(x - 405.550)_+ - (x - 505.550)_+$ |

**Table 8.** Optimal solutions under different risk measures with the constraint of $\mathscr{F}_2$ under the settings that $X \sim F(x) = \exp\left\{-\left(\frac{x-5}{50}\right)^{-3}\right\}$, $\theta = 3$, and $K = 120$.

| $\alpha$ | The Risk Measure $\rho$ | $\rho_\alpha(T_{I^*}(X))$ | $I^*(x)$ |
|---|---|---|---|
| 0.90 | VaR | 100.108 | $(x - 80.741)_+ - (x - 110.863)_+$ |
| | LVaR($\omega = 0.2$) | 111.032 | $(x - 80.741)_+ - (x - 110.863)_+$ |
| | LVaR($\omega = 0.5$) | 123.111 | $(x - 80.741)_+ - (x - 237.333)_+$ |
| | LVaR($\omega = 0.8$) | 126.578 | $(x - 80.741)_+ - (x - 237.333)_+$ |
| | TVaR | 128.889 | $(x - 80.741)_+ - (x - 237.333)_+$ |
| 0.95 | VaR | 108.290 | $(x - 80.741)_+ - (x - 139.571)_+$ |
| | LVaR($\omega = 0.2$) | 121.955 | $(x - 80.741)_+ - (x - r)_+$ with $r \in [80.741, 237.333]$ |
| | LVaR($\omega = 0.5$) | 128.889 | $(x - 80.741)_+ - (x - 237.333)_+$ |
| | LVaR($\omega = 0.8$) | 135.822 | $(x - 80.741)_+ - (x - 237.333)_+$ |
| | TVaR | 140.444 | $(x - 80.741)_+ - (x - 237.333)_+$ |
| 0.97 | VaR | 112.261 | $(x - 80.741)_+ - (x - 165.102)_+$ |
| | LVaR($\omega = 0.2$) | 125.037 | $(x - 80.741)_+ - (x - 237.333)_+$ |
| | LVaR($\omega = 0.5$) | 136.593 | $(x - 80.741)_+ - (x - 237.333)_+$ |
| | LVaR($\omega = 0.8$) | 148.148 | $(x - 80.741)_+ - (x - 237.333)_+$ |
| | TVaR | 155.852 | $(x - 80.741)_+ - (x - 237.333)_+$ |
| 0.99 | VaR | 117.308 | $(x - 80.741)_+ - (x - 236.691)_+$ |
| | LVaR($\omega = 0.2$) | 140.444 | $(x - 80.741)_+ - (x - 237.333)_+$ |
| | LVaR($\omega = 0.5$) | 175.111 | $(x - 80.741)_+ - (x - 237.333)_+$ |
| | LVaR($\omega = 0.8$) | 209.778 | $(x - 80.741)_+ - (x - 237.333)_+$ |
| | TVaR | 232.889 | $(x - 80.741)_+ - (x - 237.333)_+$ |
| 0.999 | VaR | 384.917 | $(x - r)_+ - (x - \beta(r))_+$ with $r \in [0, 384.179]$ and $\beta(r) \in [237.333, 504.917]$ |
| | LVaR($\omega = 0.2$) | 434.923 | $(x - r)_+ - (x - \beta(r))_+$ with $r \in [0, 384.179]$ and $\beta(r) \in [237.333, 504.917]$ |
| | LVaR($\omega = 0.5$) | 509.933 | $(x - r)_+ - (x - \beta(r))_+$ with $r \in [0, 384.179]$ and $\beta(r) \in [237.333, 504.917]$ |
| | LVaR($\omega = 0.8$) | 584.943 | $(x - r)_+ - (x - \beta(r))_+$ with $r \in [0, 384.179]$ and $\beta(r) \in [237.333, 504.917]$ |
| | TVaR | 634.950 | $(x - r)_+ - (x - \beta(r))_+$ with $r \in [0, 384.179]$ and $\beta(r) \in [237.333, 504.917]$ |

**Table 9.** Optimal solutions under different risk measures with the constraint of $\mathscr{F}_1$.

| $\alpha$ | The Risk Measure $\rho$ | $\rho_\alpha(T_{I^*}(X))$ | $I^*(x)$ |
|---|---|---|---|
| 0.90 | VaR | 77.587 | $(x - 63.496)_+ - (x - 83.203)_+$ |
| | LVaR($\omega = 0.2$) | 86.448 | $(x - 63.496)_+ - (x - 83.203)_+$ |
| | LVaR($\omega = 0.5$) | 99.738 | $(x - 63.496)_+ - (x - r)_+$ with $r \in [83.203, 183.496]$ |
| | LVaR($\omega = 0.8$) | 102.565 | $(x - 64.314)_+ - (x - 184.314)_+$ |
| | TVaR | 104.436 | $(x - 64.863)_+ - (x - 184.863)_+$ |
| 0.95 | VaR | 85.980 | $(x - 63.496)_+ - (x - 106.736)_+$ |
| | LVaR($\omega = 0.2$) | 96.987 | $(x - 63.496)_+ - (x - 106.736)_+$ |
| | LVaR($\omega = 0.5$) | 104.436 | $(x - 64.863)_+ - (x - 184.863)_+$ |
| | LVaR($\omega = 0.8$) | 109.982 | $(x - 66.532)_+ - (x - 186.532)_+$ |
| | TVaR | 113.624 | $(x - 67.662)_+ - (x - 187.663)_+$ |
| 0.97 | VaR | 90.005 | $(x - 63.496)_+ - (x - 127.432)_+$ |
| | LVaR($\omega = 0.2$) | 101.312 | $(x - 63.949)_+ - (x - 183.949)_+$ |
| | LVaR($\omega = 0.5$) | 110.592 | $(x - 66.720)_+ - (x - 186.720)_+$ |
| | LVaR($\omega = 0.8$) | 119.598 | $(x - 69.577)_+ - (x - 189.577)_+$ |
| | TVaR | 125.452 | $(x - 71.529)_+ - (x - 191.529)_+$ |
| 0.99 | VaR | 95.128 | $(x - 65.043)_+ - (x - 185.043)_+$ |
| | LVaR($\omega = 0.2$) | 113.624 | $(x - 67.662)_+ - (x - 187.662)_+$ |
| | LVaR($\omega = 0.5$) | 139.572 | $(x - 76.565)_+ - (x - 196.566)_+$ |
| | LVaR($\omega = 0.8$) | 163.204 | $(x - 86.098)_+ - (x - 206.099)_+$ |
| | TVaR | 177.766 | $(x - 92.674)_+ - (x - 212.675)_+$ |
| 0.999 | VaR | 280.907 | $(x - 279.867)_+ - (x - 399.867)_+$ |
| | LVaR($\omega = 0.2$) | 320.917 | $(x - 279.867)_+ - (x - 399.867)_+$ |
| | LVaR($\omega = 0.5$) | 380.933 | $(x - 279.867)_+ - (x - 399.867)_+$ |
| | LVaR($\omega = 0.8$) | 440.949 | $(x - 279.867)_+ - (x - 399.867)_+$ |
| | TVaR | 480.954 | $(x - 281.130)_+ - (x - 401.130)_+$ |

Table 10. Optimal solutions under different risk measures with the constraint of $\mathscr{F}_2$.

| $\alpha$ | The Risk Measure $\rho$ | $\rho_\alpha(T_{I^*}(X))$ | $I^*(x)$ |
|---|---|---|---|
| 0.90 | VaR | 77.587 | $(x-63.496)_+ - (x-83.203)_+$ |
| | LVaR($\omega=0.2$) | 86.448 | $(x-63.496)_+ - (x-83.203)_+$ |
| | LVaR($\omega=0.5$) | 99.738 | $(x-63.496)_+ - (x-r)_+$ with $r \in [63.496, 257.326]$ |
| | LVaR($\omega=0.8$) | 101.186 | $(x-63.496)_+ - (x-257.326)_+$ |
| | TVaR | 102.151 | $(x-63.496)_+ - (x-257.326)_+$ |
| 0.95 | VaR | 85.980 | $(x-63.496)_+ - (x-106.736)_+$ |
| | LVaR($\omega=0.2$) | 96.987 | $(x-63.496)_+ - (x-257.326)_+$ |
| | LVaR($\omega=0.5$) | 102.151 | $(x-63.496)_+ - (x-257.326)_+$ |
| | LVaR($\omega=0.8$) | 105.046 | $(x-63.496)_+ - (x-257.326)_+$ |
| | TVaR | 106.976 | $(x-63.496)_+ - (x-257.326)_+$ |
| 0.97 | VaR | 90.005 | $(x-63.496)_+ - (x-127.432)_+$ |
| | LVaR($\omega=0.2$) | 100.543 | $(x-63.496)_+ - (x-257.326)_+$ |
| | LVaR($\omega=0.5$) | 105.368 | $(x-63.496)_+ - (x-257.326)_+$ |
| | LVaR($\omega=0.8$) | 110.193 | $(x-63.496)_+ - (x-257.326)_+$ |
| | TVaR | 113.410 | $(x-63.496)_+ - (x-257.326)_+$ |
| 0.99 | VaR | 95.084 | $(x-63.496)_+ - (x-185.043)_+$ |
| | LVaR($\omega=0.2$) | 106.976 | $(x-63.496)_+ - (x-257.326)_+$ |
| | LVaR($\omega=0.5$) | 121.453 | $(x-63.496)_+ - (x-257.326)_+$ |
| | LVaR($\omega=0.8$) | 135.929 | $(x-63.496)_+ - (x-257.326)_+$ |
| | TVaR | 145.580 | $(x-63.496)_+ - (x-257.326)_+$ |
| 0.999 | VaR | 239.867 | $(x-r)_+ - (x-\beta(r))_+$ with $r \in [0.243, 238.049]$ and $\beta(r) \in [257.326, 399.867]$ |
| | LVaR($\omega=0.2$) | 279.877 | $(x-r)_+ - (x-\beta(r))_+$ with $r \in [0.243, 238.049]$ and $\beta(r) \in [257.326, 399.867]$ |
| | LVaR($\omega=0.5$) | 339.893 | $(x-r)_+ - (x-\beta(r))_+$ with $r \in [0.243, 238.049]$ and $\beta(r) \in [257.326, 399.867]$ |
| | LVaR($\omega=0.8$) | 399.909 | $(x-r)_+ - (x-\beta(r))_+$ with $r \in [0.243, 238.049]$ and $\beta(r) \in [257.326, 399.867]$ |
| | TVaR | 439.920 | $(x-r)_+ - (x-\beta(r))_+$ with $r \in [0.243, 238.049]$ and $\beta(r) \in [257.326, 399.867]$ |

## 5. Conclusions

It is well known that VaR and TVaR are the most popular risk measures and both of them have been widely used in the literature (see, e.g., Cai and Tan 2007; Cai et al. 2008; Lu et al. 2016). However, these two risk measures were generally considered separately. By noticing that $\mathrm{TVaR}_\alpha \geq \mathrm{VaR}_\alpha$ and the size of capital reserves may be significantly different based on VaR and TVaR, we have proposed a new family of risk measures named LVaR which is a linear combination of VaR and TVaR. This new risk measure considers VaR and TVaR simultaneously and helps us to obtain a risk assessment by LVaR. A major issue in a reinsurance contract is to guarantee a balance between the ceded risk and the contract premium. From the perspective of the insurer, the problem is how to choose a ceded loss function such that the total losses are as small as possible (or the total benefits are as large as possible). To this end, we have revisited the optimal reinsurance problem by minimizing the LVaR of the total risk of the insurer when the reinsurer's risk exposure has an upper limit.

In this paper, the optimal reinsurance problem has been studied between one insurer and one reinsurer. However, as introduced by Boonen et al. (2021), the setting with one insurer and multiple reinsurers may be more realistic in practice. This inspires our future work to consider the case with multiple reinsurers.

The most important results are presented in Theorems 1 and 2, which provide the solutions of our optimal reinsurance model. It is shown that the two-layer reinsurance is always the optimal reinsurance strategy under two types of constraints. Furthermore, we have found that the minimums of LVaR of an insurer's total risks are larger than those of VaR and smaller than those of TVaR. The same holds for deductibles. In numerical illustrations, we have noted significant differences in the size of capital reserves depending on adopted weight coefficients, and moreover, higher capital reserves are obtained for larger weight coefficients.

By introducing the weighted coefficient $\omega$, the new risk measure LVaR may help financial institutions, such as insurance companies, to quantify risk based on different

situations since the weight coefficient $\omega$ reflects the user's attitude towards risk. More precisely, the larger $\omega$ means the higher sensitivity to the severity of losses exceeding VaR. Most importantly, if a insurer intends to transfer some part of risk to one reinsurance, our work may provide a way to determine the size of the transferred risk.

**Author Contributions:** S.N. wrote Section 4. Q.X. and Z.P. wrote the remaining sections. All authors have read and agreed to the published version of the manuscript.

**Funding:** This research received no external funding.

**Data Availability Statement:** Not applicable.

**Acknowledgments:** The authors would like to thank the Editor and the three referees for careful reading and comments which greatly improved the paper.

**Conflicts of Interest:** The authors declare no conflicts of interest.

**Appendix A**

**Proof of Lemma 2.** According to the sign of $\delta$, we consider $\delta \geq 0$ and $\delta < 0$ in turn.

In the case of $\delta \geq 0$, for any $I(x) \in \mathscr{F}_1$, let

$$g(x; a, b) = (x - e_\alpha)_+ - (x - \mathrm{VaR}_\alpha(X))_+,$$

where $e_\alpha = \mathrm{VaR}_\alpha(X) - I(\mathrm{VaR}_\alpha(X))$. Obviously, $(e_\alpha, \mathrm{VaR}_\alpha(X)) \in \mathscr{D}_1$, and hence $g(x; a, b) \in \mathscr{G}_1$. Furthermore, we have $g(x; e_\alpha, \mathrm{VaR}_\alpha(X)) = 0 \leq I(x)$ for $0 \leq x < e_\alpha$ and $g(x; e_\alpha, \mathrm{VaR}_\alpha(X)) = I(\mathrm{VaR}_\alpha(X)) \leq I(x)$ for $x \geq \mathrm{VaR}_\alpha(X)$. For $e_\alpha \leq x < \mathrm{VaR}_\alpha(X)$, by using the property of Lipschitz continuity of $I(x)$, we have $g(x; e_\alpha, \mathrm{VaR}_\alpha(X)) = x - \mathrm{VaR}_\alpha(X) + I(\mathrm{VaR}_\alpha(X)) \leq I(x)$. Thus, we conclude that $g(x; e_\alpha, \mathrm{VaR}_\alpha(X)) \leq I(x)$ for $x \geq 0$. Noting that $g(\mathrm{VaR}_\alpha(X); e_\alpha, \mathrm{VaR}_\alpha(X)) = I(\mathrm{VaR}_\alpha(X))$, by (11), we have $\mathrm{LVaR}_\alpha(T_g(X)) \leq \mathrm{LVaR}_\alpha(T_I(X))$. Hence, the desired result follows for $\delta \geq 0$.

In the case of $\delta < 0$, for any $I(x) \in \mathscr{F}_1$, let

$$g(x; a, b) = (x - e_\alpha)_+ - (x - (e_\alpha + L))_+,$$

where $e_\alpha$ is defined as above. We can easily check that $\mathrm{VaR}_\alpha(X) \leq e_\alpha + L$ and, hence, $g(x; a, b) \in \mathscr{G}_1$. We can prove that $g(x; e_\alpha, e_\alpha + L) = 0 \leq I(x)$ for $0 \leq x < e_\alpha$, and $g(x; e_\alpha, e_\alpha + L) = L \geq I(x)$ for $x \geq e_\alpha + L$. If $e_\alpha \leq x < \mathrm{VaR}_\alpha(X)$, we have from Lipschitz continuity of $I(x)$ that

$$g(x; e_\alpha, e_\alpha + L) = x - \mathrm{VaR}_\alpha(X) + I(\mathrm{VaR}_\alpha(X)) \leq I(x).$$

Similarly, if $\mathrm{VaR}_\alpha(X) \leq x < e_\alpha + L$, we have

$$g(x; e_\alpha, e_\alpha + L) = x - \mathrm{VaR}_\alpha(X) + I(\mathrm{VaR}_\alpha(X)) \geq I(x).$$

Hence, by applying (11), we have the desired result for $\delta < 0$. The proof is complete. $\square$

**Proof of Theorem 1.** By Lemma 2, we can solve the optimal problem (14) instead of (4). Applying the definition of $g(x)$ in (13), we have

$$g(\mathrm{VaR}_\alpha(X)) = \mathrm{VaR}_\alpha(X) - a, \tag{A1}$$

$$\begin{aligned}
\int_0^{\mathrm{VaR}_\alpha(X)} g(x) dF_X(x) &= \int_a^{\mathrm{VaR}_\alpha(X)} (x - a) dF_X(x) \\
&= \int_a^{\mathrm{VaR}_\alpha(X)} S_X(t) dt - (\mathrm{VaR}_\alpha(X) - a) S_X(\mathrm{VaR}_\alpha(X)), \quad \text{(A2)}
\end{aligned}$$

and

$$\int_{\text{VaR}_\alpha(X)}^\infty g(x)dF_X(x) = \int_{\text{VaR}_\alpha(X)}^b (x-a)dF_X(x) + \int_b^\infty (b-a)dF_X(x)$$

$$= \int_{\text{VaR}_\alpha(X)}^b S_X(t)dt + (\text{VaR}_\alpha(X) - a)S_X(\text{VaR}_\alpha(X)). \quad \text{(A3)}$$

Combining (A1)–(A3) with (11), we obtain

$$\begin{aligned}
\text{LVaR}_\alpha(T_g(X)) &= \omega\Psi(\alpha) + \text{VaR}_\alpha(X) - (1-\omega)(\text{VaR}_\alpha(X) - a) + (1+\theta)\int_a^{\text{VaR}_\alpha(X)} S_X(t)dt \\
&\quad - (1+\theta)(\text{VaR}_\alpha(X) - a)S_X(\text{VaR}_\alpha(X)) + \delta\int_{\text{VaR}_\alpha(X)}^b S_X(t)dt \\
&\quad + \delta(\text{VaR}_\alpha(X) - a)S_X(\text{VaR}_\alpha(X)) \\
&= \omega\Psi(\alpha) + \text{VaR}_\alpha(X) - (1-\omega)(\text{VaR}_\alpha(X) - a) + (1+\theta)\int_a^{\text{VaR}_\alpha(X)} S_X(t)dt \\
&\quad + \delta\int_{\text{VaR}_\alpha(X)}^b S_X(t)dt - \omega(\text{VaR}_\alpha(X) - a) \\
&= \omega\Psi(\alpha) + a + (1+\theta)\int_a^{\text{VaR}_\alpha(X)} S_X(t)dt + \delta\int_{\text{VaR}_\alpha(X)}^b S_X(t)dt \\
&\triangleq \phi(a,b). \quad \text{(A4)}
\end{aligned}$$

The rest of the work is to minimize (A4) according the sign of $\delta$.

(i) If $\delta < 0$ (which implies $\alpha > \theta^*$), taking the partial derivative with respect to $b$ on $\phi(a,b)$ yields

$$\frac{\partial\phi(a,b)}{\partial b} = \delta S_X(b) < 0. \quad \text{(A5)}$$

From (A5) and (12), for any $\alpha \in (p_0, 1)$ and $(a,b) \in \mathcal{D}_1$, we always have $\phi(a,b) \geq \phi(a, a+L)$. Note that $(a,b) \in \mathcal{D}_1$ implies $(a, a+L) \in \mathcal{D}_1$, so that the minimum of $\text{LVaR}_\alpha(T_g(X))$ must be attained at $(a,b)$ with $b = a+L$. As a result, it suffices to solve the following optimal problem

$$\min_{a\in[\max\{0,\text{VaR}_\alpha(X)-L\},\text{VaR}_\alpha(X)]} \varphi(a), \quad \text{(A6)}$$

where

$$\varphi(a) = \phi(a, a+L) = \omega\Psi(\alpha) + a + (1+\theta)\int_a^{\text{VaR}_\alpha(X)} S_X(t)dt + \delta\int_{\text{VaR}_\alpha(X)}^{a+L} S_X(t)dt.$$

By taking the first two derivatives of $\varphi(a)$, we have

$$\varphi'(a) = 1 - (1+\theta)S_X(a) + \delta S_X(a+L)$$

and

$$\varphi''(a) = (1+\theta)f_X(a) - \delta S_X(a+L) > 0,$$

which implies that $\varphi(a)$ is convex. In addition, we can obtain

$$\begin{aligned}
\varphi'(\text{VaR}_{\theta^*}(X)) &= 1 - (1+\theta)(1-\theta^*) + \delta S_X(\text{VaR}_{\theta^*}(X) + L) \\
&= \delta S_X(\text{VaR}_{\theta^*}(X) + L) < 0, \quad \text{(A7)}
\end{aligned}$$

and

$$
\begin{aligned}
\varphi'(\mathrm{VaR}_\alpha(X)) &= 1 - (1+\theta)(1-\alpha) + \delta S_X(\mathrm{VaR}_\alpha(X) + L) \\
&\geq 1 - (1+\theta)(1-\alpha) + \delta S_X(\mathrm{VaR}_\alpha(X)) \\
&= 1 - \omega \geq 0.
\end{aligned}
\tag{A8}
$$

Let $a_0$ be the solution to $\varphi'(a) = 0$. We consider the following two cases.

(a)  If $\mathrm{VaR}_\alpha(X) - L \leq a_0$, we have $\varphi'(\mathrm{VaR}_\alpha(X) - L) \leq 0$. Combining with (A7) and (A8), we conclude that the minimum of $\varphi(a)$ must be attained at $a_0$ with $a_0 \in (\max\{\mathrm{VaR}_{\theta^*}(X), \mathrm{VaR}_\alpha(X) - L\}, \mathrm{VaR}_\alpha(X))$.

(b)  If $\mathrm{VaR}_\alpha(X) - L > a_0$, we obtain $\varphi'(\mathrm{VaR}_\alpha(X) - L) > 0$. Then by the monotonicity of $\varphi(a)$ in $[\mathrm{VaR}_\alpha(X) - L, \mathrm{VaR}_\alpha(X)]$, we find that $\varphi(a)$ attains its minimum at $\mathrm{VaR}_\alpha(X) - L$.

Hence, the proof of (i) is complete.

(ii)  If $\delta = 0$ (which implies $\alpha \geq \theta^*$), we have

$$
\mathrm{LVaR}_\alpha(T_g(X)) = \omega \Psi(\alpha) + a + (1+\theta) \int_a^{\mathrm{VaR}_\alpha(X)} S_X(t)dt \overset{\Delta}{=} \eta(a)
$$

with $a \in [\max\{0, \mathrm{VaR}_\alpha(X) - L\}, \mathrm{VaR}_\alpha(X)]$. Taking the first derivatives of $\eta(a)$, we obtain

$$
\eta'(a) = 1 - (1+\theta)S_X(a).
$$

From the monotonicity of $\eta(a)$, we find that the minimum of $\eta(a)$ is attained at $\mathrm{VaR}_{\theta^*}(X)$ for $\mathrm{VaR}_\alpha(X) - L \leq \mathrm{VaR}_{\theta^*}(X)$, as well as at $\mathrm{VaR}_\alpha(X) - L$ for $\mathrm{VaR}_\alpha(X) - L > \mathrm{VaR}_{\theta^*}(X)$. Equivalently, if $\mathrm{VaR}_\alpha(X) - L \leq \mathrm{VaR}_{\theta^*}(X)$, then $\mathrm{LVaR}_\alpha(T_I(X))$ attains its minimum at $I^*(x) = (x - \mathrm{VaR}_{\theta^*}(X))_+ - (x - r)_+$, where $r$ is any real number in $[\mathrm{VaR}_\alpha(X), \mathrm{VaR}_{\theta^*}(X) + L]$. If $\mathrm{VaR}_\alpha(X) - L > \mathrm{VaR}_{\theta^*}(X)$, $\mathrm{LVaR}_\alpha(T_I(X))$ attains its minimum at $I^*(x) = (x - (\mathrm{VaR}_\alpha(X) - L))_+ - (x - \mathrm{VaR}_\alpha(X))_+$.

(iii)  If $\delta > 0$, taking the partial derivative of $\phi(a, b)$ with respect to $b$ yields

$$
\frac{\partial \phi(a, b)}{\partial b} = \delta S_X(b) > 0.
\tag{A9}
$$

From (A9) and the properties of $\mathscr{D}_1$, for any $\alpha \in (p_0, 1)$ and $(a, b) \in \mathscr{D}_1$, we have $\phi(a, b) \geq \phi(a, \mathrm{VaR}_\alpha(X))$. Thus, the minimum of $\mathrm{LVaR}_\alpha(T_I(X))$ must be attained at $(a, \mathrm{VaR}_\alpha(X)) \in \mathscr{D}_1$. Consequently, it suffices to solve the following optimal problem

$$
\min_{a \in [\max\{0, \mathrm{VaR}_\alpha(X) - L\}, \mathrm{VaR}_\alpha(X)]} \gamma(a),
$$

where

$$
\gamma(a) = \omega \Psi(\alpha) + a + (1+\theta) \int_a^{\mathrm{VaR}_\alpha(X)} S_X(t)dt.
$$

Taking the first derivatives of $\gamma(a)$ yields

$$
\gamma'(a) = 1 - (1+\theta)S_X(a).
\tag{A10}
$$

We consider the following cases.

(a)  If $\alpha \leq \theta^*$, then from (A10), the minimum of $\mathrm{LVaR}_\alpha(T_I(X))$ is attained at $I^*(x) = 0$.

(b)  If $\alpha > \theta^*$ and $\mathrm{VaR}_\alpha(X) - L \leq \mathrm{VaR}_{\theta^*}(X)$, from (A10), $\mathrm{LVaR}_\alpha(T_I(X))$ attains its minimum at $I^*(x) = (x - \mathrm{VaR}_{\theta^*}(X))_+ - (x - \mathrm{VaR}_\alpha(X))_+$.

(c)  If $\alpha > \theta^*$ and $\mathrm{VaR}_\alpha(X) - L > \mathrm{VaR}_{\theta^*}(X)$, then the minimum of $\mathrm{LVaR}_\alpha(T_I(X))$ is attained at $I^*(x) = (x - (\mathrm{VaR}_\alpha(X) - L))_+ - (x - \mathrm{VaR}_\alpha(X))_+$.

The proof is complete.    □

**Proof of Lemma 3.** For any $I(x) \in \mathscr{F}_2$, let $g_1(x) = (x - e_\alpha)_+ - (x - \mathrm{VaR}_\alpha(X))_+$ with $e_\alpha = \mathrm{VaR}_\alpha(X) - I(\mathrm{VaR}_\alpha(X))$, and $g_2(x) = (x - e_\alpha)_+ - (x - e_\alpha - M)_+$ and $g_3(x) = x - (x - M)_+$ with $M = K + (1 + \theta) \mathbb{E}[I(X)]$. By similar arguments to Lemma 2, we can show that $g_1(x) \le I(x) \le g_3(x)$ and $\mathbb{E}[g_1(X)] \le \mathbb{E}[I(X)] \le \mathbb{E}[g_3(X)]$.

We consider the following two cases, $\mathbb{E}[g_2(X)] \le \mathbb{E}[I(X)]$ and $\mathbb{E}[g_2(X)] > \mathbb{E}[I(X)]$. In the case of $\mathbb{E}[g_2(X)] \le \mathbb{E}[I(X)]$, for any function $g_4(x; a) = (x - a)_+ - (x - a - M)_+$, we have

$$\mathbb{E}[g_4(X; a)] = \int_a^{a+M} S_X(x)dx.$$

Thus,

$$\frac{\partial \mathbb{E}[g_4(X; a)]}{\partial a} = S_X(a + M) - S_X(a) < 0,$$

which indicates that $\mathbb{E}[g_4(X; a)]$ is decreasing with respect to $a$. From the continuity of $\mathbb{E}[g_4(X; a)]$ with respect to $a$, there must exist $g(x; c) = (x - c)_+ - (x - c - M)_+$ with $0 \le c \le e_\alpha$, such that $\mathbb{E}[g(X)] = \mathbb{E}[I(X)]$ and $g_2(x) \le g(x) \le g_3(x)$. We can easily verify that $g(x) \in \mathscr{G}_2$ for $0 \le c \le e_\alpha$ and $g(x) \ge g_2(x) \ge I(x)$ for $x \ge \mathrm{VaR}_\alpha(X)$. Then, by (10), $g(x)$ is the desired function, that is, $\mathrm{LVaR}_\alpha(T_g(X)) \le \mathrm{LVaR}_\alpha(T_I(X))$.

Similarly, in the case of $\mathbb{E}[g_2(X)] > \mathbb{E}[I(X)]$, for any function $g_5(x; b) = (x - e_\alpha)_+ - (x - b)_+$, we obtain that

$$\mathbb{E}[g_5(X; a)] = \int_{e_\alpha}^b S_X(x)dx$$

and

$$\frac{\partial \mathbb{E}[g_5(X; a)]}{\partial b} = S_X(b) > 0,$$

so that $\mathbb{E}[g_5(X; a)]$ is increasing with respect to $b$. Because $\mathbb{E}[g_5(X; a)]$ is continuous with respect to $a$, there must exist $g(x; k) = (x - e_\alpha)_+ - (x - k)_+$ with $k \in [\mathrm{VaR}_\alpha(X), e_\alpha + M)$, such that $\mathbb{E}[g(X)] = \mathbb{E}[I(X)]$ and $g_1(x) \le g(x) < g_2(x)$. As a result, we have $g(\mathrm{VaR}_\alpha(X)) \ge g_1(\mathrm{VaR}_\alpha(X)) = I(\mathrm{VaR}_\alpha(X))$. If $x \in [0, \mathrm{VaR}_\alpha(X)]$, we can easily check that $g(x) \le I(x)$, and combining this with $\mathbb{E}[g(X)] = \mathbb{E}[I(X)]$, we obtain $\int_{\mathrm{VaR}_\alpha(X)}^\infty g(x)dF_X(x) \ge \int_{\mathrm{VaR}_\alpha(X)}^\infty I(x)dF_X(x)$. Furthermore, for $k \in [\mathrm{VaR}_\alpha(X), e_\alpha + M)$, we can find that $k - e_\alpha < M = K + (1 + \theta) \mathbb{E}[I(X)] = K + (1 + \theta) \mathbb{E}[g(X)]$, which implies that $g(x) \in \mathscr{G}_2$. Hence, by (10), we have $\mathrm{LVaR}_\alpha(T_g(X)) \le \mathrm{LVaR}_\alpha(T_I(X))$ and the proof is complete.    □

**Proof of Theorem 2.** For any $g(x; a, b) \in \mathscr{G}_2$, if $b \le \mathrm{VaR}_\alpha(X)$, we have $g(\mathrm{VaR}_\alpha(X)) = b - a$ and

$$\int_{\mathrm{VaR}_\alpha(X)}^\infty g(x)dF_X(x) = \int_{\mathrm{VaR}_\alpha(X)}^\infty (b - a)dF_X(x) = (b - a)(1 - \alpha).$$

Then, applying (10), we obtain that

$$
\begin{aligned}
\mathrm{LVaR}_\alpha(T_g(X)) &= \omega \Psi(\alpha) + \mathrm{VaR}_\alpha(X) - (1 - \omega)(b - a) + (1 + \theta) \mathbb{E}[g(X)] - \omega(b - a) \\
&= \omega \Psi(\alpha) + \mathrm{VaR}_\alpha(X) - [b - a - (1 + \theta) \mathbb{E}[g(X)]] \\
&= \omega \Psi(\alpha) + \mathrm{VaR}_\alpha(X) - \psi(a, b) \\
&\overset{\triangle}{=} \eta_1(a, b)
\end{aligned}
\tag{A11}
$$

for $b \le \mathrm{VaR}_\alpha(X)$, where $\psi(a, b) = b - a - (1 + \theta) \mathbb{E}[g(X)]$. If $b > \mathrm{VaR}_\alpha(X)$, we have that $g(\mathrm{VaR}_\alpha(X)) = \mathrm{VaR}_\alpha(X) - a$ and

$$
\begin{aligned}
\int_{\mathrm{VaR}_\alpha(X)}^{\infty} g(x)dF_X(x) &= \int_{\mathrm{VaR}_\alpha(X)}^{b}(x-a)dF_X(x) + \int_{b}^{\infty}(b-a)dF_X(x) \\
&= \int_{\mathrm{VaR}_\alpha(X)}^{b}\int_{a}^{x}dt dF_X(x) + (b-a)S_X(x) \\
&= \int_{a}^{\mathrm{VaR}_\alpha(X)}\int_{\mathrm{VaR}_\alpha(X)}^{b}dF_X(x)dt + \int_{\mathrm{VaR}_\alpha(X)}^{b}\int_{t}^{b}dF_X(x)d + (b-a)S_X(x) \\
&= \int_{a}^{\mathrm{VaR}_\alpha(X)}S_X(\mathrm{VaR}_\alpha(X)) - S_X(b)dt + \int_{\mathrm{VaR}_\alpha(X)}^{b}S_X(t) - S_X(b)dt + (b-a)S_X(x) \\
&= \int_{\mathrm{VaR}_\alpha(X)}^{b}S_X(t)dt + (\mathrm{VaR}_\alpha(X) - a)(1-\alpha).
\end{aligned}
$$

Let $h(b) \overset{\Delta}{=} b - \frac{\omega}{1-\alpha}\int_{\mathrm{VaR}_\alpha(X)}^{b}S_X(t)dt$. By (10), we obtain that

$$
\begin{aligned}
\mathrm{LVaR}_\alpha\big(T_g(X)\big) &= \omega\Psi(\alpha) + \mathrm{VaR}_\alpha(X) - (1-\omega)(\mathrm{VaR}_\alpha(X) - a) + (1+\theta)\,\mathbb{E}[g(X)] \\
&\quad - \frac{\omega}{1-\alpha}\int_{\mathrm{VaR}_\alpha(X)}^{b}S_X(t)dt - \omega(\mathrm{VaR}_\alpha(X) - a) \\
&= \omega\Psi(\alpha) + a + (1+\theta)\,\mathbb{E}[g(X)] - \frac{\omega}{1-\alpha}\int_{\mathrm{VaR}_\alpha(X)}^{b}S_X(t)dt \\
&= \omega\Psi(\alpha) - [b - a - (1+\theta)\,\mathbb{E}[g(X)]] + b - \frac{\omega}{1-\alpha}\int_{\mathrm{VaR}_\alpha(X)}^{b}S_X(t)dt \\
&= \omega\Psi(\alpha) - \psi(a,b) + h(b) \\
&\overset{\Delta}{=} \eta_2(a,b). &&\text{(A12)}
\end{aligned}
$$

Thus,

$$
\mathrm{LVaR}_\alpha\big(T_g(X)\big) = \begin{cases} \eta_1(a,b), & b \le \mathrm{VaR}_\alpha(X), \\ \eta_2(a,b), & b > \mathrm{VaR}_\alpha(X). \end{cases} \tag{A13}
$$

From (A11) and (A12), we have

$$
\frac{\partial\eta_1(a,b)}{\partial a} = \frac{\partial\eta_2(a,b)}{\partial a} = 1 - (1+\theta)S_X(a), \tag{A14}
$$

$$
\frac{\partial\eta_1(a,b)}{\partial b} = -1 + (1+\theta)S_X(b), \tag{A15}
$$

$$
\frac{\partial\eta_2(a,b)}{\partial b} = \delta S_X(b) \tag{A16}
$$

and

$$
h'(b) = 1 - \frac{\omega}{1-\alpha}S_X(b). \tag{A17}
$$

For convenience, define

$$
\mathscr{D}_3 \overset{\Delta}{=} \{(a,b) \mid (a,b) \in \mathscr{D}_0, b \le \mathrm{VaR}_{\theta^*}(X)\},
$$

$$
\mathscr{D}_4 \overset{\Delta}{=} \{(a,b) \mid (a,b) \in \mathscr{D}_0, b \le \mathrm{VaR}_\alpha(X)\}.
$$

The rest is to show the results, in turn, according to $\delta < 0, \delta = 0$ and $\delta > 0$. First, we consider the case of $\delta < 0$. If $b_0^* \ge \mathrm{VaR}_{\theta^*}(X)$, note that $\delta < 0$ implies $\alpha > \theta^*$. We have from (A14)–(A17) that for any $(a,b) \in \mathscr{D}_3$,

$$
\eta_1(a,b) \ge \eta_1(a,a) = \eta_1(\mathrm{VaR}_{\theta^*}(X), \mathrm{VaR}_{\theta^*}(X))
$$

$$
\begin{aligned}
> \quad & \eta_1(\text{VaR}_{\theta^*}(X), \text{VaR}_\alpha(X)) = \eta_2(\text{VaR}_{\theta^*}(X), \text{VaR}_\alpha(X)) \\
\geq \quad & \eta_2(\text{VaR}_{\theta^*}(X), b_0^*) = \omega\Psi(\alpha) - K + h(\beta(b_0^*)).
\end{aligned}
$$

For any $(a,b) \in \mathscr{D}_4 \backslash \mathscr{D}_3$, we have

$$
\begin{aligned}
\eta_1(a,b) \quad \geq \quad & \eta_1(a, \text{VaR}_\alpha(X)) = \eta_2(a, \text{VaR}_\alpha(X)) \\
\geq \quad & \eta_2(\text{VaR}_{\theta^*}(X), \text{VaR}_\alpha(X)) \\
\geq \quad & \eta_2(\text{VaR}_{\theta^*}(X), b_0^*) = \omega\Psi(\alpha) - K + h(\beta(b_0^*)).
\end{aligned}
$$

For any $(a,b) \in \mathscr{D}_0 \backslash \mathscr{D}_4$, we obtain

$$
\begin{aligned}
\eta_2(a,b) \quad \geq \quad & \eta_2(a, \beta(a)) = \omega\Psi(\alpha) - K + h(\beta(a)) \\
\geq \quad & \omega\Psi(\alpha) - K + h(\beta(b_0^*)).
\end{aligned}
$$

Then, the minimum of $\text{LVaR}_\alpha(T_I(X))$ is $\omega\Psi(\alpha) - K + h(\beta(b_0^*))$ which is attained at $I^*(x) = (x - \text{VaR}_{\theta^*}(X))_+ - (x - b_0^*)_+$.

Similarly, if $b_0^* < \text{VaR}_{\theta^*}(X)$, for any $(a,b) \in \mathscr{D}_3$,

$$
\begin{aligned}
\eta_1(a,b) \quad \geq \quad & \eta_1(a,a) = \eta_1(\text{VaR}_{\theta^*}(X), \text{VaR}_{\theta^*}(X)) \\
\geq \quad & \eta_1(\text{VaR}_{\theta^*}(X), b_0^*) = \omega\Psi(\alpha) + \text{VaR}_\alpha(X) - K.
\end{aligned}
$$

For any $(a,b) \in \mathscr{D}_4 \backslash \mathscr{D}_3$, we obtain

$$
\begin{aligned}
\eta_1(a,b) \quad \geq \quad & \eta_1(a, \min\{\beta(a), \text{VaR}_\alpha(X)\}) \\
\geq \quad & \eta_1(r, \beta(r)) = \omega\Psi(\alpha) + \text{VaR}_\alpha(X) - K,
\end{aligned}
$$

where $r$ is any real number in $[\max\{0, a_0^*\}, a_1^*]$. For any $(a,b) \in \mathscr{D}_0 \backslash \mathscr{D}_4$, we have

$$
\begin{aligned}
\eta_2(a,b) \quad \geq \quad & \eta_2(a, \beta(a)) = \omega\Psi(\alpha) - K + h(\beta(a)) \\
> \quad & \omega\Psi(\alpha) - K + h(\text{VaR}_\alpha(X)) = \omega\Psi(\alpha) + \text{VaR}_\alpha(X) - K.
\end{aligned}
$$

Hence, the minimum of $\text{LVaR}_\alpha(T_I(X))$ is attained at $I^*(x) = (x - r)_+ - (x - \beta(r))_+$ with $r \in [\max\{0, a_0^*\}, a_1^*]$ as $b_0^* < \text{VaR}_\alpha(X)$. The proof for the case of $\delta < 0$ is complete.

For the case of $\delta = 0$, note that $\delta = 0$ implies $\alpha \geq \theta^*$. If $\alpha = \theta^*$, we have $b_0^* \geq \text{VaR}_{\theta^*}(X) = \text{VaR}_\alpha(X)$, then for any $(a,b) \in \mathscr{D}_4$,

$$
\begin{aligned}
\eta_1(a,b) \quad \geq \quad & \eta_1(a,a) \\
= \quad & \eta_1(\text{VaR}_\alpha(X), \text{VaR}_\alpha(X)) \\
= \quad & \eta_2(\text{VaR}_\alpha(X), \text{VaR}_\alpha(X)) \\
= \quad & \omega\Psi(\alpha) + \text{VaR}_\alpha(X).
\end{aligned}
$$

For $(a,b) \in \mathscr{D}_0 \backslash \mathscr{D}_4$,

$$
\begin{aligned}
\eta_2(a,b) \quad \geq \quad & \eta_2(\text{VaR}_{\theta^*}(X), b) \\
= \quad & \eta_2(\text{VaR}_\alpha, b) \\
= \quad & \omega\Psi(\alpha) + \text{VaR}_\alpha(X).
\end{aligned}
$$

Hence, the minimum of $\text{LVaR}_\alpha(T_I(X))$ is $\omega\Psi(\alpha) + \text{VaR}_\alpha(X)$ which is attained at $I^*(x) = (x - \text{VaR}_\alpha(X))_+ - (x - r)_+$ with $r \in [\text{VaR}_\alpha(X), \beta(\text{VaR}_\alpha(X))]$.

If $\alpha > \theta^*$ and $b_0^* \geq \text{VaR}_\alpha(X)$, for any $(a,b) \in \mathscr{D}_3$,

$$
\begin{aligned}
\eta_1(a,b) \quad \geq \quad & \eta_1(a,a) = \eta_1(\text{VaR}_{\theta^*}(X), \text{VaR}_{\theta^*}(X)) \\
> \quad & \eta_1(\text{VaR}_{\theta^*}(X), \text{VaR}_\alpha(X)) \\
= \quad & \eta_2(\text{VaR}_{\theta^*}(X), \text{VaR}_\alpha(X))
\end{aligned}
$$

$$= \omega\Psi(\alpha) + \mathrm{VaR}_{\theta^*}(X) + (1+\theta)\int_{\mathrm{VaR}_{\theta^*}(X)}^{\mathrm{VaR}_\alpha(X)} S_X(x)dx.$$

For any $(a,b) \in \mathscr{D}_4 \backslash \mathscr{D}_3$,

$$
\begin{aligned}
\eta_1(a,b) &\geq \eta_1(a,\mathrm{VaR}_\alpha(X)) = \eta_2(a,\mathrm{VaR}_\alpha(X)) \\
&\geq \eta_2(\mathrm{VaR}_{\theta^*}(X),\mathrm{VaR}_\alpha(X)) \\
&= \eta_2(\mathrm{VaR}_{\theta^*}(X),r) \\
&= \omega\Psi(\alpha) + \mathrm{VaR}_{\theta^*}(X) + (1+\theta)\int_{\mathrm{VaR}_{\theta^*}(X)}^{\mathrm{VaR}_\alpha(X)} S_X(x)dx,
\end{aligned}
$$

where $r$ is any real number in $[\mathrm{VaR}_{\theta^*}(X),\mathrm{VaR}_\alpha(X)]$. For any $(a,b) \in \mathscr{D}_0 \backslash \mathscr{D}_4$,

$$
\begin{aligned}
\eta_2(a,b) &\geq \eta_2(\mathrm{VaR}_{\theta^*}(X),b) \\
&= \eta_2(\mathrm{VaR}_{\theta^*}(X),r) \\
&= \omega\Psi(\alpha) + \mathrm{VaR}_{\theta^*}(X) + (1+\theta)\int_{\mathrm{VaR}_{\theta^*}(X)}^{\mathrm{VaR}_\alpha(X)} S_X(x)dx,
\end{aligned}
$$

where $r$ is any real number in $[\mathrm{VaR}_\alpha(X),b_0^*]$. Hence, the minimum of $\mathrm{LVaR}_\alpha(T_I(X))$ is

$$\omega\Psi(\alpha) + \mathrm{VaR}_{\theta^*}(X) + (1+\theta)\int_{\mathrm{VaR}_{\theta^*}(X)}^{\mathrm{VaR}_\alpha(X)} S_X(x)dx$$

and is attained at $I^*(x) = (x - \mathrm{VaR}_{\theta^*}(X))_+ - (x-r)_+$, where $r$ is any real number in $(\mathrm{VaR}_{\theta^*}(X),b_0^*]$.

If $\alpha > \theta^*$ and $b_0^* < \mathrm{VaR}_\alpha(X)$, for any $(a,b) \in \mathscr{D}_3$,

$$
\begin{aligned}
\eta_1(a,b) &\geq \eta_1(a,a) = \eta_1(\mathrm{VaR}_{\theta^*}(X),\mathrm{VaR}_{\theta^*}(X)) \\
&> \eta_1(\mathrm{VaR}_{\theta^*}(X),b_0^*) \\
&= \omega\Psi(\alpha) + \mathrm{VaR}_\alpha(X) - K.
\end{aligned}
$$

For any $(a,b) \in \mathscr{D}_4 \backslash \mathscr{D}_3$,

$$
\begin{aligned}
\eta_1(a,b) &\geq \eta_1(a,\min\{\beta(a),\mathrm{VaR}_\alpha(X)\}) \\
&\geq \eta_1(r,\beta(r)) \\
&= \omega\Psi(\alpha) + \mathrm{VaR}_\alpha(X) - K,
\end{aligned}
$$

where $r$ is any real number in $[\max\{0,a_0^*\},a_1^*]$. For any $(a,b) \in \mathscr{D}_0 \backslash \mathscr{D}_4$,

$$
\begin{aligned}
\eta_2(a,b) &= \eta_2(a,\beta(a)) \\
&= \omega\Psi(\alpha) - K + h(\beta(a)) \\
&> \omega\Psi(\alpha) - K + h(\beta(\mathrm{VaR}_\alpha(X))) \\
&= \omega\Psi(\alpha) + \mathrm{VaR}_\alpha(X) - K.
\end{aligned}
$$

Hence, the minimum of $\mathrm{LVaR}_\alpha(T_I(X))$ is $\omega\Psi(\alpha) + \mathrm{VaR}_\alpha(X) - K$ which is attained at $I^*(x) = (x-r)_+ - (x-\beta(r))_+$, where $r$ is any real number in $[\max\{0,a_0^*\},a_1^*]$.

For the case of $\delta > 0$, if $\alpha \leq \theta^*$, for any $(a,b) \in \mathscr{D}_4$,

$$\eta_1(a,b) \geq \eta_1(a,a) = \omega\Psi(\alpha) + \mathrm{VaR}_\alpha(X).$$

For any $(a,b) \in \mathscr{D}_0 \backslash \mathscr{D}_4$,

$$\eta_2(a,b) \geq \eta_2(a,\mathrm{VaR}_\alpha(X)) \geq \eta_2(\mathrm{VaR}_\alpha(X),\mathrm{VaR}_\alpha(X)) = \omega\Psi(\alpha) + \mathrm{VaR}_\alpha(X).$$

Hence, the minimum of $LVaR_\alpha(T_I(X))$ is $\omega\Psi(\alpha) + VaR_\alpha(X)$ as $\alpha \leq \theta^*$ and is attained at $I^*(x) = 0$.

If $\alpha > \theta^*$ and $b_0^* \geq VaR_\alpha(X)$, for any $(a,b) \in \mathscr{D}_3$,

$$
\begin{aligned}
\eta_1(a,b) &\geq \eta_1(a,a) = \eta_1(VaR_{\theta^*}(X), VaR_{\theta^*}(X)) \\
&> \eta_1(VaR_{\theta^*}(X), VaR_\alpha(X)) \\
&= \omega\Psi(\alpha) + VaR_{\theta^*}(X) + (1+\theta)\int_{VaR_{\theta^*}(X)}^{VaR_\alpha(X)} S_X(x)dx.
\end{aligned}
$$

For any $(a,b) \in \mathscr{D}_4 \backslash \mathscr{D}_3$,

$$
\begin{aligned}
\eta_1(a,b) &\geq \eta_1(a, VaR_\alpha(X)) \\
&\geq \eta_1(VaR_{\theta^*}(X), VaR_\alpha(X)) \\
&= \omega\Psi(\alpha) + VaR_{\theta^*}(X) + (1+\theta)\int_{VaR_{\theta^*}(X)}^{VaR_\alpha(X)} S_X(x)dx.
\end{aligned}
$$

For any $(a,b) \in \mathscr{D}_0 \backslash \mathscr{D}_4$,

$$
\begin{aligned}
\eta_2(a,b) &\geq \eta_2(VaR_{\theta^*}(X), b) \\
&\quad \eta_2(VaR_{\theta^*}(X), VaR_\alpha(X)) \\
&= \eta_1(VaR_{\theta^*}(X), VaR_\alpha(X)) \\
&= \omega\Psi(\alpha) + VaR_{\theta^*}(X) + (1+\theta)\int_{VaR_{\theta^*}(X)}^{VaR_\alpha(X)} S_X(x)dx.
\end{aligned}
$$

Hence, the minimum of $LVaR_\alpha(T_I(X))$ is

$$
\omega\Psi(\alpha) + VaR_{\theta^*}(X) + (1+\theta)\int_{VaR_{\theta^*}(X)}^{VaR_\alpha(X)} S_X(x)dx
$$

and is attained at $I^*(x) = (x - VaR_{\theta^*}(X))_+ - (x - VaR_\alpha(X))_+$.

Finally, if $\alpha > \theta^*$ and $b_0^* > VaR_\alpha(X)$, for any $(a,b) \in \mathscr{D}_3$,

$$
\begin{aligned}
\eta_1(a,b) &\geq \eta_1(a,a) = \eta_1(VaR_{\theta^*}(X), VaR_{\theta^*}(X)) \\
&> \eta_1(VaR_{\theta^*}(X), b_0^*) \\
&= \omega\Psi(\alpha) + VaR_\alpha(X) - K.
\end{aligned}
$$

For any $(a,b) \in \mathscr{D}_4 \backslash \mathscr{D}_3$,

$$
\begin{aligned}
\eta_1(a,b) &\geq \eta_1(a, \min\{\beta(a), VaR_\alpha(X)\}) \\
&\geq \eta_1(r, \beta(r)) \\
&= \omega\Psi(\alpha) + VaR_\alpha(X) - K,
\end{aligned}
$$

where $r$ is any real number in $[\max\{0, a_0^*\}, a_1^*]$.

For any $(a,b) \in \mathscr{D}_0 \backslash \mathscr{D}_4$,

$$
\begin{aligned}
\eta_2(a,b) &> \eta_2(a, VaR_\alpha(X)) = \eta_1(a, VaR_\alpha(X)) \\
&\geq \eta_1(VaR_{\theta^*}(X), VaR_\alpha(X)) \\
&= \omega\Psi(\alpha) + VaR_\alpha(X) - \psi(VaR_{\theta^*}(X), VaR_\alpha(X)) \\
&> \omega\Psi(\alpha) + VaR_\alpha(X) - K.
\end{aligned}
$$

Hence, the minimum of $LVaR_\alpha(T_I(X))$ is $\omega\Psi(\alpha) + VaR_\alpha(X) - K$ and is attained at $I^*(x) = (x-r)_+ - (x - \beta(r))_+$, where $r$ is any real number in $[\max\{0, a_0^*\}, a_1^*]$.

Therefore, by combining the above analyses, we complete the proof of Theorem 2. □

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
