# Peer review of "Optimal Reinsurance under the Linear Combination of Risk Measures in the Presence of Reinsurance Loss Limit"

_risks, doi:10.3390/risks11070125_

Round 1

Reviewer 1 Report

See my report.

Reviewer 2 Report

I am pleased to have the opportunity to review this research paper. This study attempted to explore the Optimal reinsurance under the linear combination of risk measures in the presence of reinsurance loss limit. Although the topic of this research study is interesting and fits within the journal scope, I think authors should apply the comments indicated below to increase the quality of research justification, contributions and findings. The manuscript know lacks in scientific style and structure.

First of all, paper research gap. Please improve this part in introduction section. Introduction is very general and lacked alignment to the research findings, no discussion was provided to derive the implication from. Theoretical and pragmatics implication are vague and need to be better aligned with this paper theoretical underpinnings and proposed process. Furthermore, there is insufficient support and weak arguments in support of the objective that is proposed as well as the model developed. In the final part of the introduction the objectives proposed, originality and gap that would be better covered. Also how the author will perform the methodology.

the topic of this research study is interesting and fits within the journal scope, I think authors should apply the comments indicated to increase the quality of research justification, contributions and findings

What is the originality of this research?  Paper research gap and originality should be better presented at the end of introduction section

Please consider this structure for manuscript final part.

-Discussion

-Conclusion

-Managerial Implication

-Practical/Social Implications

-Discussion needs to be a coherent and cohesive set of arguments that take us beyond this study in particular, and help us see the relevance of what authors have proposed. Authors should create an independent “Discussion” section. Author need to contextualize the findings in the literature, and need to be explicit about the added value of your study towards that literature. Also other studies should be cited to increase the theoretical background of each of the method used. Findings should be contextualized in the literature and should be explicit about the added value of the study towards the literature. Limitations and future research

Questions to be answered:

What practical/professional and academic consequences will this study have for the future of scientific literature (theoretical contributions)?

Why is this study necessary? should make clear arguments to explain what is the originality and value of the proposed model. This should be stated in the final paragraphs of introduction and conclusion sections.

Reviewer 3 Report

The quality of english writing is sufficient, and it only requires some minor corrections to typos and imprecisions.

Round 2

Reviewer 1 Report

All the comments in the former report were addressed in the revised version.

Reviewer 2 Report

congratulations, your work is now better. But, before being accepted, I ask that you better substantiate the need for your study, better demonstrate the gap, and clearly explain the contribution of your study to academia and companies.
